# Detecting global and local hierarchical structures in cell-cell communication using CrossChat

Xinyi Wang[1], Axel A. Almet [1,2] ✉ & Qing Nie [1,2,3] ✉

Cell-cell communication (CCC) occurs across different biological scales, ranging from interactions between large groups of cells to interactions between individual cells, forming a hierarchical structure. Globally, CCC may exist between clusters or only subgroups of a cluster with varying size, while locally, a group of cells as sender or receiver may exhibit distinct signaling properties. Current existing methods infer CCC from single-cell RNA-seq or Spatial Transcriptomics only between predefined cell groups, neglecting the existing hierarchical structure within CCC that are determined by signaling molecules, in particular, ligands and receptors. Here, we develop CrossChat, a novel computational framework designed to infer and analyze the hierarchical cell-cell communication structures using two complementary approaches: a global hierarchical structure using a multi-resolution clustering method, and multiple local hierarchical structures using a tree detection method. This framework provides a comprehensive approach to understand the hierarchical relationships within CCC that govern complex tissue functions. By applying our method to two nonspatial scRNA-seq datasets sampled from COVID-19 patients and mouse embryonic skin, and two spatial transcriptomics datasets generated from Stereo-seq of mouse embryo and 10x Visium of mouse wounded skin, we showcase CrossChat's functionalities for analyzing both global and local hierarchical structures within cell-cell communication.

In a multicellular organism, biological functions are performed by cells coordinating via cell-cell communication (CCC). Cells communicate with each other by secreting molecular signals, which are received by nearby cells. CCC instructs crucial cellular functions, including cell development, homeostasis, and immune responses in disease[1–4]. CCC occurs at different biological scales. While some communications occur over a wide range of cells, others may occur only among a small subset of cells. The first example of such a hierarchy in CCC is related to cell types at varying scales. Specifically, there may be sub-cell types within a cell type. T follicular helper cells, which are a subset of CD4 + T cells, have been found to secrete IL-4 and IL-21 ligands, which are received by B cells to promote cell proliferation and B cell

differentiation into plasma B cells or germinal center B cells[5]. Another ligand-receptor interaction, CD40L-CD40 interaction occurs at a wider range of cells, from CD4 + T cells to B cells. This interaction is crucial for T-cell-dependent B cell proliferation and differentiation[6]. These two examples of ligand-receptor interactions demonstrate the possibility of a hierarchy among different CCC activities. The second example of a possible hierarchy in CCC is determined completely by the cells where ligands/receptors involved in the interaction are expressed. For example, there are genes expressed during a specific phase of a cell cycle among many different cell types, and they can perform important biological functions such as regulating cell growth, DNA replication, and cell division[7]. As a specific example, senescent

[1]Department of Mathematics, University of California, Irvine, CA, USA. [2]The NSF-Simons Center for Multiscale Cell Fate Research, University of California, Irvine, CA, USA. [3]Department of Developmental and Cell Biology, University of California, Irvine, CA, USA. ✉e-mail: aalmet@uci.edu; qnie@uci.edu

cells can induce paracrine effects on other within their tissue micro-environment via the senescence-associated secretory phenotype, which is characterized by the secretion of a wide array of inflammatory cytokines, chemokines, and growth factors[8]. Therefore, considering cells that express a certain ligand/receptor, while relaxing the restriction that they should belong to a certain cell type, is biologically more relevant when studying CCC.

Therefore, identifying which groups of cells exhibit hierarchical CCC structure could increase one's understanding of the structured biological functions that occur among cell groups due to inherent structures within ligand-receptor interactions. Such an understanding of structures within CCC will shed light on the systematic mechanism of cellular events, such as cell development and differentiation, as well as orchestration of different functions across various biological conditions such as inflammation, wound healing, and cancer.

Recent advances in single-cell RNA-sequencing and spatial transcriptomics technologies have paved the way to understanding CCC, and a rich number of methods have been developed to infer CCC based on scRNA-seq or spatial transcriptomics data[9–24]. Current tools for modeling CCC typically operate under specific assumptions and focus on interactions among groups of cells that are predefined. However, this approach has two significant drawbacks. Firstly, it overlooks the intricate cell structures shaped by ligands and receptors, crucial components in CCC. More importantly, it only accounts for CCC between cell groups at a one level of clustering, which overlook the varying scales of cell states, i.e. there may be substates within clusters, and further substates within those substates. In the example of the interaction between T follicular helper cells and B cells through secretion of IL-4 and IL-12 ligands, if we only consider one scale of clustering, where T cells are a cluster, the two interactions sent from T follicular helper cells, and CD4 + T cells respectively, will both be assigned as interactions from T cells. A coarse scale of clustering may not be able to ascribe the interactions to the correct cell types as precisely as desired. However, focusing solely on the finest scale of clusters may cause us to miss the broader "big picture" functions of T cells and B cells. Therefore, it is essential to consider the cellular hierarchy rather than just a single level of clusters.

The key to overcoming these limitations is to develop a method that can analyze CCC between cell groups at multiple clustering resolutions defined by expression of signal ligands and receptors. Hierarchies in cell groups occur throughout biology. For example, a hierarchy naturally develops during cell differentiation, when a sub-cluster of cells start to perform more specialized functions among a larger group of cells. Consequently, we can define hierarchical structures within CCC with respect to two complementary perspectives. First, from a global perspective, the hierarchical structure divides all cells into clusters at different scales, where a cell group at a coarser scale indicates a general cell group, and a cell group at a finer scale indicates a more differentiated subcluster. A ligand-receptor interaction can be specific to a pair of cell groups belonging to any scale along this hierarchy, and thus the global hierarchy of CCC can be induced from this hierarchy of cells. Second, from a local perspective, cells expressing a ligand A may be a subset of cells expressing ligand B (with a similar analogy for signal receptors). If we analyze ligands and receptors with respect to their expressing cell groups, some ligands or receptors will form natural tree structures based on the overlap or lack thereof between the cell groups that express these ligands. Several recent studies investigated the existing hierarchical patterns in scRNA-seq datasets[25–29]. However, they focus on the hierarchy from a global perspective, and do not study the local hierarchy with respect to CCC.

Thus, we propose CrossChat, which detects and analyzes both global and local hierarchical structures within CCC generated by ligand-receptor interactions using single-cell and spatial transcriptomics. CrossChat consists of two core methods, CrossChatH and CrossChatT, and their respective extensions to spatial transcriptomics

data, CrossChatH-S and CrossChatT-S. CrossChatH detects a global hierarchical communication structure based on a hierarchical community detection method. CrossChatT is able to detect multiple local hierarchical structures based with respect to ligand gene expression or receptor gene expression using a tree detection method. By incorporating spatial information, both methods can be easily adapted to spatial datasets, namely CrossChatH-S and CrossChatT-S. CrossChat presents several major advantages. First, it detects structures within CCC that do not rely on predefined cell type annotations. Second, it detects the structure of CCC with respect to ligand and receptor gene expression, and thus more accurately reflects the inherent structures within CCC. Third, it provides a comprehensive view on hierarchical structures within CCC with respect to both global and local perspectives. To the best of our knowledge, CrossChat is the first method to detect and analyze hierarchical structures within CCC. CrossChat is available as an open-source Python package, that provides inference, visualization, and downstream applications of hierarchical structures within cell-cell communications.

## Results
### Overview of CrossChat
From a global perspective, hierarchical structures CCC can be uncovered by hierarchically clustering cells according to their gene expression similarity or their similarity in ligand or receptor gene expression. Some ligand-receptor interactions may occur between smaller cell groups in which cells are highly similar, while other interactions may occur between larger cell groups where cells are less similar (Fig. 1a). From a local perspective, hierarchical structures in CCC can be uncovered by analyzing, for example, whether a group of cells that secrete a signal ligand contains a smaller subset of cells that secrete a secondary signal ligand. Similar structures may exist with respect to signal receptors or ligand-receptor pairs (Fig. 1b). Given these hierarchical structures within ligands or receptors, we can build hierarchical relations between ligand-receptor interactions (Fig. 1c). These hierarchical structures of ligands, receptors, or interactions can be visualized using trees (Fig. 1d–f).

CrossChatH investigates the global hierarchical structure within CCC based on hierarchical clustering of cells. It first detects hierarchical cell groups based on gene expression or ligands/receptors expression similarity, and then calculates CCC activity between clusters (Fig. 2a). The input to the method either a log-normalized gene expression matrix or a raw count matrix. For a raw count matrix input, it will first log-normalize it. It then calculates a cell-cell similarity graph based on cosine similarity, which is then used to produce a K-nearest neighbor graph, where the nodes represent the cells, and the edge weights represent the similarity of cells. Using PyGenStability[30], a multi-resolution community detection method with generalized Markov Stability[31], we obtain a hierarchy structure within cells. Next, we obtain an ordered list of specific ligand-receptor pairs, ranked by their specificity to ligand and receptor clusters in the hierarchy. For each ligand-receptor pair, CCC is calculated using CellChat[9]. We tested the robustness of hierarchical clustering of CrossChatH to noise, sparsity, and normalization, selection of K in K-nearest neighbor graph construction (Supplementary Fig. 1a–d).

CrossChatT investigates the local hierarchical structure with respect to ligands, receptors, or ligand-receptor interactions using Bron-Kerbosch, a graph search algorithm[32] (Fig. 2a, Supplementary Fig. 2). The input to the method is a raw gene expression matrix, which is then subsetted to consider only ligand and receptor genes and then binarized to construct a support matrix. For the set of ligands (receptors), we connect any pair of ligands (receptors) based on whether they adhere to the principle of being disjoint or inclusive. This procedure generates a gene-gene relationship graph where graph nodes represent genes, and edges represent gene-gene relationships. The Bron-Kerbosch algorithm is used on this gene relationship graph

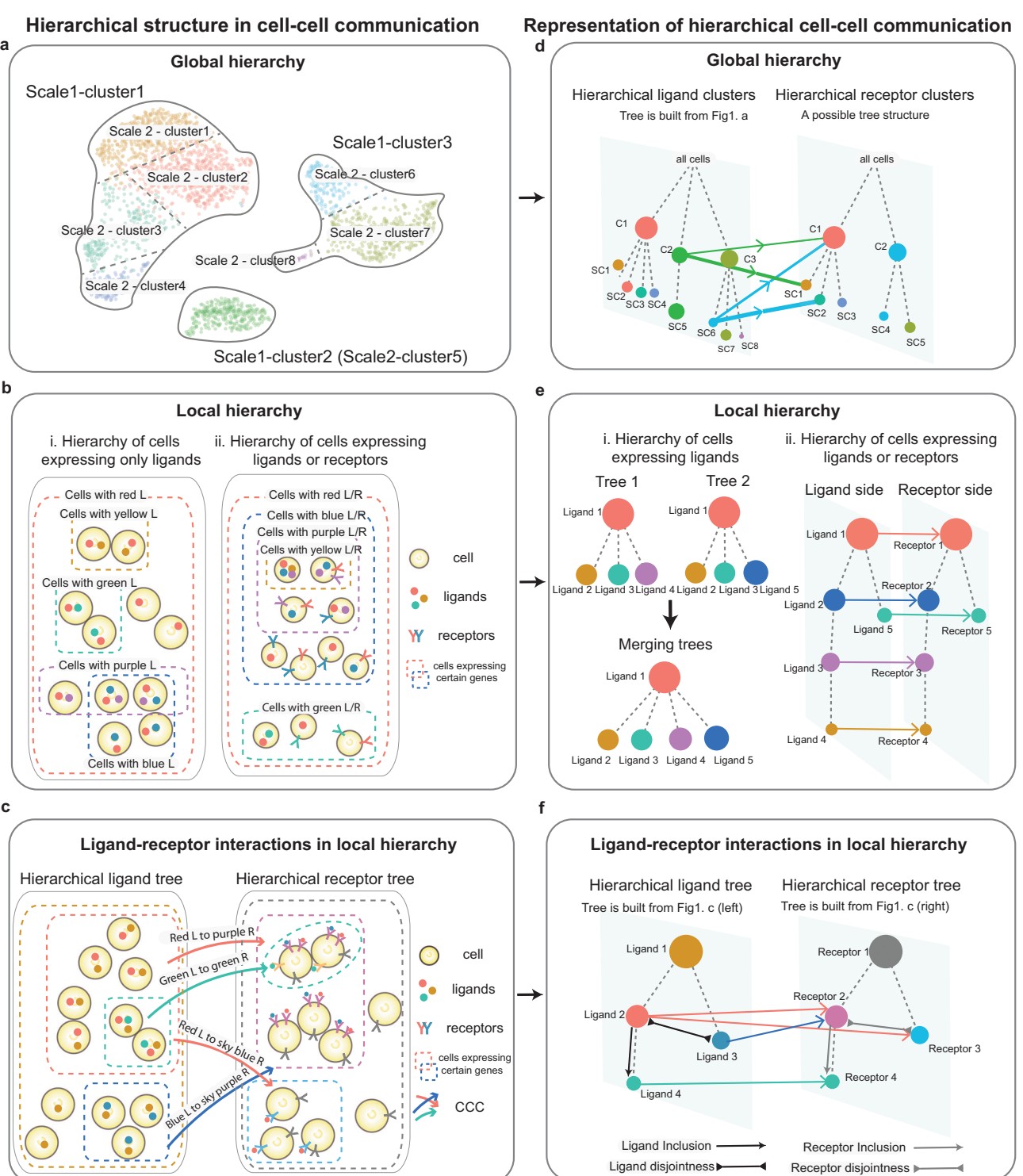

**Fig. 1 | Hierarchical structures in cell-cell communications. a** A hierarchical clustering of cells. Cell groups at scale 2 are subclusters of cell groups at scale 1. **b** Hierarchical structures within ligands, receptors, or union of ligands and receptors. (i) In the hierarchy of ligands, all cells express red ligands, some the cells express yellow ligands, and some cells express green ligands, etc. (ii) In the hierarchy of ligand-receptor unions, all cells express either the red ligand or receptor. Within the group of cells expressing the yellow ligand or receptor, there are also cells expressing either the purple ligand or receptor, and blue ligand or receptor.

**c** Hierarchical structures of ligand-receptor interactions. There are four hierarchical relationships of ligand-receptor interactions ligand inclusion, ligand, disjointness, receptor inclusion, and receptor disjointness. **d** Representations of cell-cell communication between hierarchical clustering of cells based on ligands (left) and receptors (right). The node sizes correspond to the number of cells in the groups, and the edge widths represent interaction strengths. **e** Tree representations of hierarchical structures within ligands, receptors, or ligand-receptor unions. **f** Representations of hierarchical relations between ligand-receptor interactions.

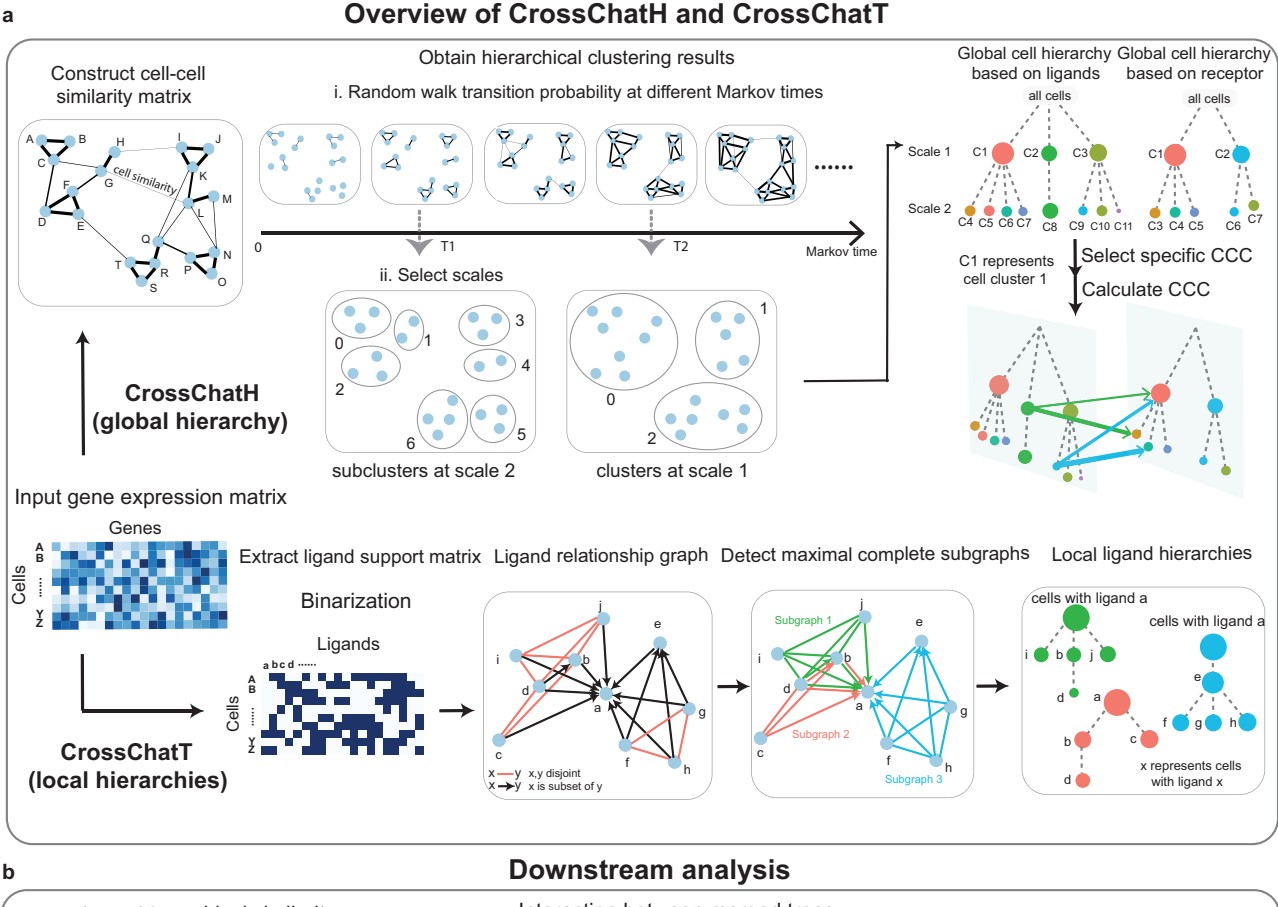

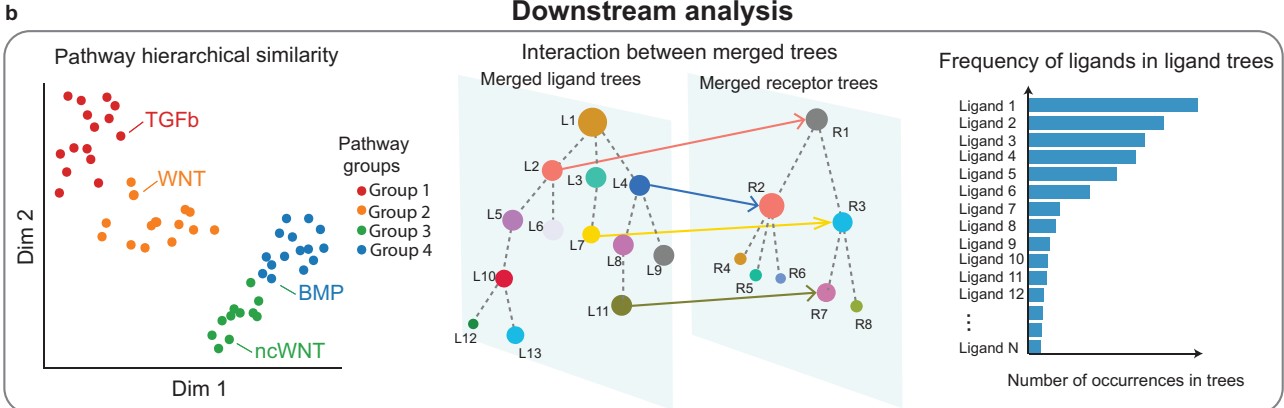

**Fig. 2 | Overview of CrossChat. a** Method overview of CrossChatH and CrossChatT. (i) CrossChatH: hierarchical clustering. The input is a gene expression matrix. CrossChatH calculates a cell-cell similarity matrix and uses a random walk-based method on the similarity matrix to obtain hierarchical clustering results. After obtaining ligand clusters and receptor clusters, it runs CellChat to calculate cell-cell communication between clusters. (ii) CrossChatT: tree detection of ligand-receptor interactions. The input is a gene expression matrix, which is then binarized to form a ligands/receptors support matrix. A gene-gene relationship graph is constructed, where gene pairs whose support is either disjoint or inclusive are connected. The Bron-Kerbosch algorithm is used on the gene relationship graph to find maximal complete subgraphs. Each subgraph represents a tree structure of ligands/receptors. **b** Downstream analysis of CrossChatH and CrossChatT: (i) Similarity of pathways based on ligand/receptor distributions over hierarchical clusters. (ii) Visualization of ligand-receptor interactions between union of ligand/receptors trees. (iii) Frequency of ligands/receptors in all ligand/receptor trees.

to find maximal complete subgraphs (largest subgraph in which each pair of nodes is connected). Each subgraph represents a local hierarchical structure within the set of ligands/receptors, where each pair of ligands/receptors are either disjoint or inclusive, forming a tree structure. For each pair of ligand tree and receptor tree which has at least one known ligand-receptor interaction, we calculate CCC using CellChat. We tested how different binarization thresholds could affect the detection of local hierarchies (Supplementary Fig. 1e–h). We found that the number of detected ligand trees decrease as we increased the binarization threshold (Supplementary Fig. 1e). However, we observed

that neither the number of detected receptor trees (Supplementary Fig. 1f) nor the average number of ligands/receptors in ligands/receptors trees (Supplementary Fig. 1g–h) change significantly as the binarization threshold changes.

CrossChat provides functionality for downstream analysis of these hierarchical structures extracted from CCC (Fig. 2c). First, CrossChat groups signaling pathways or ligand-receptor pairs based on their hierarchical distribution of ligands/receptors by calculating the cosine similarity between ligands/receptors distribution over hierarchical clusters, which is weighted by Markov times (see

Methods). Second, CrossChat provides a global view of interactions between ligands trees and receptor trees by visualizing interactions between a combination of ligands/receptors trees. Third, CrossChat can measure the importance of a ligand or receptor in the hierarchy of ligands/receptors by ordering them by their frequency of occurrence within the sets of ligands/receptors trees.

## Validation of CrossChat using simulated dataset and COVID-19 dataset

To benchmark the ability of CrossChatH to detect specific CCC, we simulated a dataset with 1000 cells, and 10,000 genes, among which 1000 are marker genes, and three clustering scales. In each scale, there are two clusters, four clusters, and eight clusters respectively. We reason that, as more marker genes will be inferred for larger clusters, we assign 200, 100, and 50 marker genes, respectively, to each cluster from the coarsest scale to the finest scale (Fig. 3a). There are 400, 400, and 200 marker genes in total for each scale of clusters. The details of data simulation can be found in Methods section (Synthetic data generation).

We apply CrossChatH to the log-normalized simulated count matrix to assess its ability to detect the correct marker genes of clusters at different scales. Among the top 1000 selected differentially expressed genes, we found that 789 of them are our assigned marker genes and are accurately assigned to its differential cluster, reaching a success rate of 78.9%. Indeed, at the two finer scales, the success rate reaches 86.5% and 86.0% respectively. Genes that are not ground truth marker genes tend to be assigned to larger clusters than smaller clusters, so it is expected that clusters at coarser scales will have lower success rates. As we increase the number of selected differentially expressed genes from 1000 to 2000, the percentage of correctly identified differentially expressed genes increases to 91.8%, where the success rate at scales 1, 2, and 3 reaches 96.5%, 95.0%, and 86.2%, respectively (Fig. 3b). In addition, CrossChat almost perfectly identifies multi-resolution clusters at the simulated scales 1, 2, 3, achieving an Adjusted Rand Index (ARI) of 0.99, 1, and 1, respectively, where an ARI of 0 indicates no overlap while an ARI of 1 indicates complete overlap (Fig. 3c).

The ability of CrossChatH to detect hierarchical marker genes across scales demonstrates that CrossChatH can find hierarchical structures in cells when using all genes as input. CCC can then be visualized between these hierarchical clusters. When using ligands/receptors for clustering, CrossChatH can detect hierarchical clusters that have statistically significant CCC patterns. The detected markers for each cluster in the hierarchy of cells are ligands/receptors, so it can be interpreted as a hierarchical CCC.

Furthermore, we validated the detected clusters using PBMC (peripheral blood mononuclear cells) data from COVID-19 patients. Two independent hierarchical clustering procedures are performed based on ligands only and receptors only. The boxplot shows the distribution of distance of CCC pattern of a certain cluster with the rest of the clusters. We observe that cell-cell communication patterns are robust to subsampling within each hierarchical ligands/receptors group (Fig. 3d).

The algorithm used by CrossChatT to find all maximal complete subgraphs is Bron-Kerbosch[32]. Finding hierarchical ligands/receptors trees is equivalent to finding maximal complete subgraphs in the graph of ligands/receptors that we constructed (see "CrossChatT: CCC Tree search" in Methods). To validate CrossChatT, we generated 1000 cells that expressed 100 genes. We distributed the gene expression counts so that there are at least five trees as shown in the figure (Fig. 3e). There are 15 genes generating these five trees. For each of the other 85 genes, we randomly chose 500 cells to express them. We ran the experiment 1000 times and found that CrossChatT is able to detect all five trees we generated in each trial (Fig. 3e).

## CrossChatH identifies hierarchical clusters and specific interactions in COVID-19 patients

We demonstrate the functionalities of CrossChatH on a scRNA-seq dataset of COVID-19 patients. This dataset is clustered into eight cell types, including B cells, plasma B cells, CD4 cells, CD8 cells, NK cells, dendritic cells (DC), monocytes, and megakaryocytes (Fig. 4a). Multi-resolution clustering shows that there are three clustering scales (Fig. 4a). At each cluster, we assigned each cluster to a cell type based on their cell type composition (Supplementary Fig. 3a). At scale 1, the coarsest scale, there are two clusters, where one contains a mixture of B cells and T cells, two major types of lymphocytes, and the other cluster contains monocytes. At scale 2, CrossChatH further clusters the lymphocyte cluster into B cells and two groups of T cells. At scale 3, the finest scale, finer subclusters are detected. Monocytes are further clustered into two subclusters. B cells are clustered into three subclusters, identifying plasma B cells as a cell subtype. T cells are clustered into three subclusters, where CD4 and CD8 cells are identified cell subtypes. We note that megakaryocytes are merged into the T cell clusters. This is because megakaryocytes are scarce in number, and the community detection algorithm may not be robust at identifying small-number clusters.

CrossChatH detected 102 specific ligand-receptor pairs in this dataset (the top 15 pairs are presented in Supplementary Fig. 3b). These identified CCCs generate a multi-resolution nature. For example, the ligand-receptor interaction CCL3-CCR1 interaction only occurs among a subset of monocytes, mono2 cells, and monocytes (Fig. 4b). Interactions via RETN-TLR4 only occur between monocytes and another subset of monocytes, mono1 (Fig. 4b). The interactions between the identified clusters account for most major interactions in the dataset (Fig. 4b). We also use CrossChatH to visualize CCC strength between clusters at multiple scales (Fig. 4c). These findings are consistent with previous studies, and further provide new insights into COVID-19. For example, one study showed that CCL3 is elevated in patients with more severe COVID-19 symptoms in monocytes sampled from PBMCs[33]. We observe that the specificity of CCL3 expression further suggests that CCL3 may be released by only a subset of monocytes in patients with severe COVID-19. Also, it has been shown that targeting CCL3-CCR1 may suppress "hyper activation" of the immune system in patients with COVID-19[34]. We found that CCL3 ligand and CCR1 receptor are only expressed in specific cell subtypes, suggesting that treatments should target a specific subset of monocytes as senders of CCL3 and all monocytes as receivers to suppress immune hyperactivation. A previous study also showed that activation of TLR4 is closely related to inflammation and immune hyperactivation, suggesting that targeting TLR4 could effectively reduce inflammation in severe COVID-19 patients[35]. As we found that TLR4 is most specific to mono1 cells, this suggests that treatments should specifically target a subset of monocytes as a treatment to inflammation.

Furthermore, CrossChatH can quantify the similarity of ligand-receptor interactions or pathways based on their ligand (receptor) gene distribution over hierarchical clusters, and then group them based on their similarity (See Methods). Hierarchically grouping the top 15 specific ligand-receptor pairs yielded three clusters (Fig. 4d). Group 1 represents interactions within T cells, including CCL5-ACKR2, CCL5-CCR4, GZMA-F2R (Supplementary Fig. 3b). Group 2 represents interactions from monocytes to B cells, including LGALS9-IGHM and APP-CD74. Group 3 represents interactions whose receivers are monocytes, including CCL5-CCR1, GRN-SORT1, ANXA1-FPR1.

In addition to analyzing interaction patterns based on hierarchical clusters found through all genes, CrossChatH also allows to analyze hierarchical interactions between clusters found with respect to signal ligands or signal receptors only. By neglecting the influence of genes unrelated to ligand-receptor interactions, we generate hierarchical clusters that are potentially more aligned with distributions of ligands or receptors. Applying CrossChatH clustering on all ligands and all

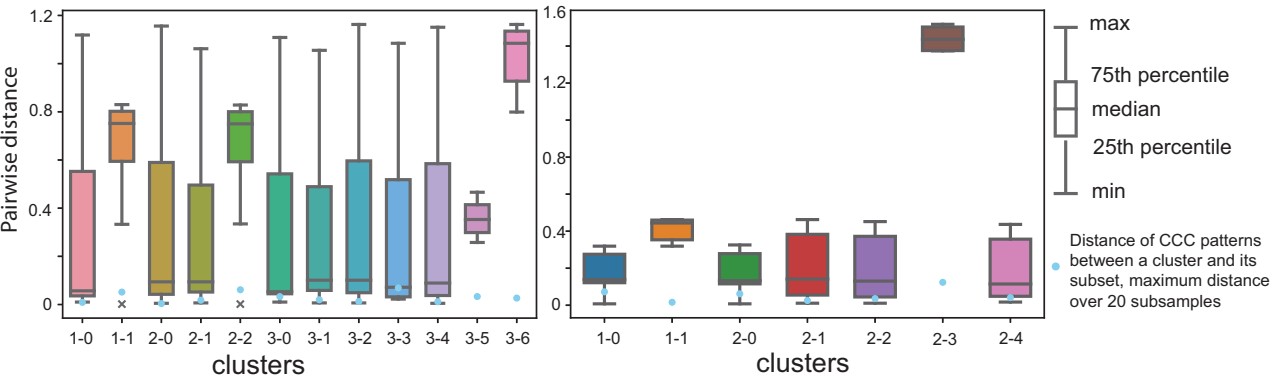

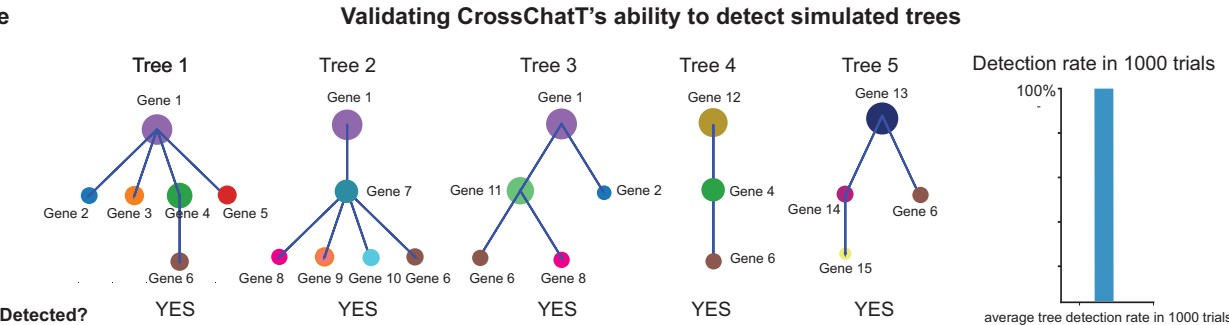

**Fig. 3 | Validation of CrossChatH. a** Synthetic data preparation. Differentially expressed genes (DEGs) are assigned to each cluster. **b** Proportion of DEGs identified at scale 1, scale 2, scale 3, and across all scales. **c** The hierarchical clustering results of CrossChatH on synthetic data. Clustering at each scale aligns well with the original hierarchical cell group assignments. **d** Validation of CrossChatH clusters using PBMC data from COVID-19 patients. Two independent hierarchical clustering procedures are performed based on ligands only and receptors only. Boxplot shows the distribution of distance of CCC pattern of a certain cluster with the rest of the clusters. Cell-cell communication patterns are robust to subsampling within each hierarchical ligands/receptors group. **e** Validation of CrossChatT to detect simulated trees. CrossChatT is capable of detecting all simulated trees in 1000 trials of simulation.

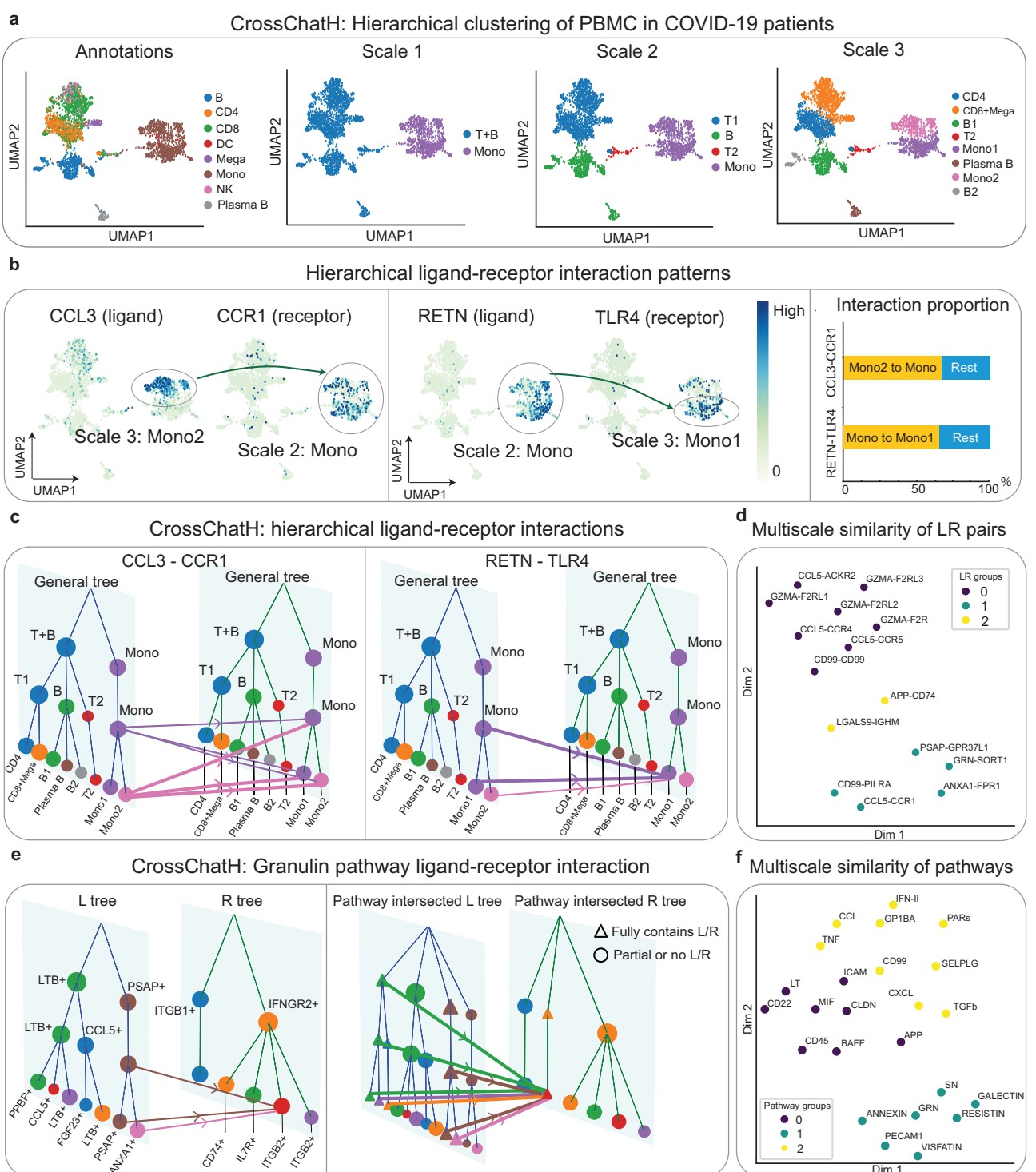

Fig. 4 | Applications of CrossChatH to scRNA-seq of PBMCs sampled from COVID-19 patients. **a** Annotations and CrossChatH hierarchical clustering results of a scRNA-seq dataset of PBMC cells from COVID-19 patients. Hierarchical clustering detects three scales of clusters, and clusters at each scale align with certain biological clusters. **b** Two ligand-receptor interactions that are specific to hierarchical clusters. CCL3 ligand is specific to Mono2, a subset of Mono, and CCR1 receptor is specific to Mono. RETN ligand is specific to Mono, and TLR4 receptor is specific to Mono1, another subset of Mono. Interaction proportion shows that the assigned cluster pairs account for most interactions through the CCL3-CCR1 and RETN-TLR4 interaction pairs. **c** Hierarchical visualization of the two ligand-receptor interactions where ligand (receptor) clusters are from CrossChatH clustering based on ligands (receptors) only. **d** Applying UMAP to top 15 specific ligand-receptor pairs based on the similarity of ligand/receptor distribution over hierarchical clusters to visualize the interaction similarity in 2D. **e** Hierarchical visualization of cell-cell communications of Granulin pathway. Left side is the visualization of interactions between hierarchical clusters generated based on ligands (receptors) only. The right side uses the support of ligands (receptors) calculate interactions with each hierarchical cluster, so each hierarchical cluster expresses a subset of cells where every cell expresses the ligands (receptors) and the other subset of cells express no ligands (receptors). **f** Applying UMAP to signaling pathways based on the similarity of ligands/receptors distribution over hierarchical clusters to visualize the interaction similarity in 2D. Source data are provided as a Source Data file.

receptors respectively gives different hierarchical clusters (Supplementary Fig. 3c–d). We assign cluster names based on their marker ligands/receptors. By observing the clustering results, the clusters based on all ligands are very similar to hierarchical clusters based on all genes, while the clusters found using all receptors are less aligned (Supplementary Fig. 3c–d). CrosschatH provides two ways to visualize the pathway interactions between ligand and receptor trees: 1) by considering all original clusters with respect to ligands and receptors or 2) constructing a "pathway intersected" tree by focusing on a specific ligand cluster and all other cells (Fig. 4e). As an example, we analyzed the Granulin pathway. While the interactions in Granulin pathway mostly occur between monocytes (PSAP is a marker ligand of this cluster), the intersected version shows that small portion of cells in T cells and B cells also secrete PSAP. Granulin has a central role in regulating the host response during infection. Its interaction with SORT1 can influence its levels and activity[36–38] and is hence seen as a promising therapeutic target for immune diseases. Our finding suggests that treatments should target monocytes. Furthermore, we clustered pathways based on their ligands and receptors distribution over hierarchical ligand clusters and receptor clusters (See Methods). Applying hierarchical grouping yields three pathway clusters (Fig. 4f). Pathways in group 1 are associated with CD8T and megakaryocytes cells and their subsets as signal senders, including CXCL, Insullin-like growth factor 2 (IFN-II), Protease-activated receptors (PARs), TGF-β, and others. Pathways in group 2 are associated with all other T cells and B cells as signal senders, including Lymphotoxin (LT), CD45, Macrophage migration inhibitory factor (MIF), B-cell activating factor (BAFF), and others. Pathways in group 3 are associated with interactions involving monocytes, including Granulin (GRN), Synuclein (SN), RESISTIN, VISFATIN, and others. Collectively, CrossChatH detects hierarchical structures of cell states based on either all genes, or ligands/receptors, and identifies specific ligand-receptor interactions in hierarchical cell structures of a given scRNA-seq dataset.

## CrossChatT identifies local hierarchical signaling structures driven by ligand-receptor interactions in mouse embryonic skin

While CrossChatH can analyze global hierarchical patterns in CCC, CrossChatT can provide a more specialized "local" perspective to analyze hierarchical structures within ligand-receptor interactions with respect to ligand or receptor gene expression. For example, in a mouse embryonic skin data, cells expresing ligand *Ocln* are a subset of cells expressing ligand *Igf2*, and cells expressing *Itga6+Itgb4* (receptor associated with *Ocln*) are disjoint with *Ocln*, which is the receptor of itself (Fig. 5a). The inclusive or disjoint relationship implicate hierarchical functions of cell groups, and such relationship of cells under CCC give rise to tree structures of ligands, receptors, and ligand-receptor interactions.

Applying CrossChatT on mouse embryonic skin E14 data[39], we found 142 hierarchical structures within ligands, and 11 hierarchical structures within receptors (Fig. 5b). There are 58 ligands in total across the 142 ligand tree structures, and 18 receptors across all receptor tree structures, indicating many overlaps between these ligand/receptor trees. There are significantly more hierarchical structures among ligands than among receptors, indicating a more structured function of cells in sending ligands during mouse skin development. Many ligand trees begin with *Igf2* as the root node. The signal ligand *Igf2* is indeed expressed in most of the cells (Fig. 5a). *Igf2* is known to primarily regulate mitogenic functions and play an important role in cell growth and development[40]. Cells expressing *Bmp5*, *Inhbb*, *Cdh1*, *Cdh5* form mutually disjoint groups (Supplementary Fig. 3a–b). *Bmp5* is mostly expressed in a subset of fibroblasts (FIB-A) and pericytes (Pericyte). In embryonic mouse skin, *Bmp5* is involved in the control of epidermal homeostasis, growth of hair follicles, and melanogenesis[41]. *Inhbb* is mainly expressed in fibroblasts (Supplementary Fig. 3a–b). It is a member of the TGF-β superfamily, and

regulates the development of the skin[42]. *Cdh1* is abundant in basal (Basal and Basal-P) and spinous cells (Spinous), which are both found in the epidermis of the skin. This gene is crucial for maintaining the structure of the epidermis by ensuring that cells adhere to each other to form a functional barrier[43]. *Cdh5* is mainly expressed in endothelial cells (ENDO) and a minor subset of FIB-A cells (Supplementary Fig. 3a–b). It ensures the integrity of blood vessels, which is important to the supply of nutrients and oxygen to the skin and removal of waste products[44]. Therefore, these signal-expressing groups, which are form disjoint subsets of the cells expressing *Igf2*, may be performing distinct functions in mouse embryonic skin. Furthermore, *Ocln* is mostly expressed in spinous cells (Spinous), a subcluster of the cell group expressing *Cdh1*, and is known to play a role in the formation and regulation of tight junctions[45,46]. *Ocln* and *Cdh1* are both important to epidermis structure, where functions of *Ocln* are more specific to spinous cells. Analyzing the hierarchical relationship of ligands/receptors in this way provides a deeper and more structured understanding of the relations between different signaling functions of cell groups.

We then investigated the interactions between ligands hierarchies and receptors hierarchies (Fig. 5c). In interaction tree 1, the adhesion gene, *Ocln*, is expressed by spinous cells, a group of differentiated keratinocytes. *Ocln-Ocln* interactions between spinous cells help form tight junctions. The ligand *Igf2* is more widely expressed, and its interaction with *Itga6_Itgb4* receptor, which is expressed in basal cells, may help promote their continued proliferation. In interaction tree 2, there exists a pair of hierarchical interactions within the WNT signaling pathway, which is pivotal to skin developmental processes including cell proliferation and differentiation[47]. *Wnt4* is expressed widely by epidermal basal cells and spinous cells, as well as dermis-resident dendritic cells and fibroblast*s, while Wnt3a* is mainly expressed by basal cells. Since *Wnt3a* is known to inhibit cell proliferation in keratinocytes, while *Wnt4* promotes cell proliferation in general[48], we may infer that *Wnt3a* ligand binding competes with the binding of *Wnt4*, regulating cell proliferation in basal cells.

CrossChatT can analyze multiple ligand trees (or receptor trees) in tandem (Fig. 5d), facilitating an understanding of the relationship within ligands (receptors). We note that these "aggregated" ligand trees (or receptor trees) do not necessarily exhibit the disjoint relationships between ligands in the individual trees. To measure the importance of ligands/receptors in their hierarchical structures, CrossChatT measures the frequency of the ligand/receptor occurrence across all ligands/receptors trees (Fig. 5e). *Igf2* is ranked first due to its wide expression by cells. While *Ptprf* is ranked first among receptors due to its wide expression, *Ocln*, which is more specifically expressed in spinous cells, is also ranked highly as its expression is disjoint from many other receptors. In addition to detecting hierarchical ligands/receptors trees, CrossChatT is also capable of detecting trees within cells that are involved in the ligand-receptor interactions (Fig. 5f). Tree 1 is associated with the WNT signaling pathway, while Tree 2 is associated with the MK signaling pathway.

## CrossChatS reveals hierarchical clusters and ligand-receptor interactions in spatial datasets

CrossChat can also be applied to spatial transcriptomics data by incorporating spatial information in both CrossChatH and CrossChatT. We demonstrate utility of CrossChatH-S to a Stereo-seq data of a mouse embryo on day E16.5[49]. We compared the hierarchical clustering results between two trials: one trial use spatial information, and the other trial does not use spatial information (Supplementary Fig. 4a, b). We can see that when we incorporate spatial locations by concatenating the spatial vectors to PCA embeddings, the clusters detected tend to be more spatially coherent. For example, the neurons are further segmented into multiple spatial regions. We also observed that the multiscale clusters generated with spatial information have

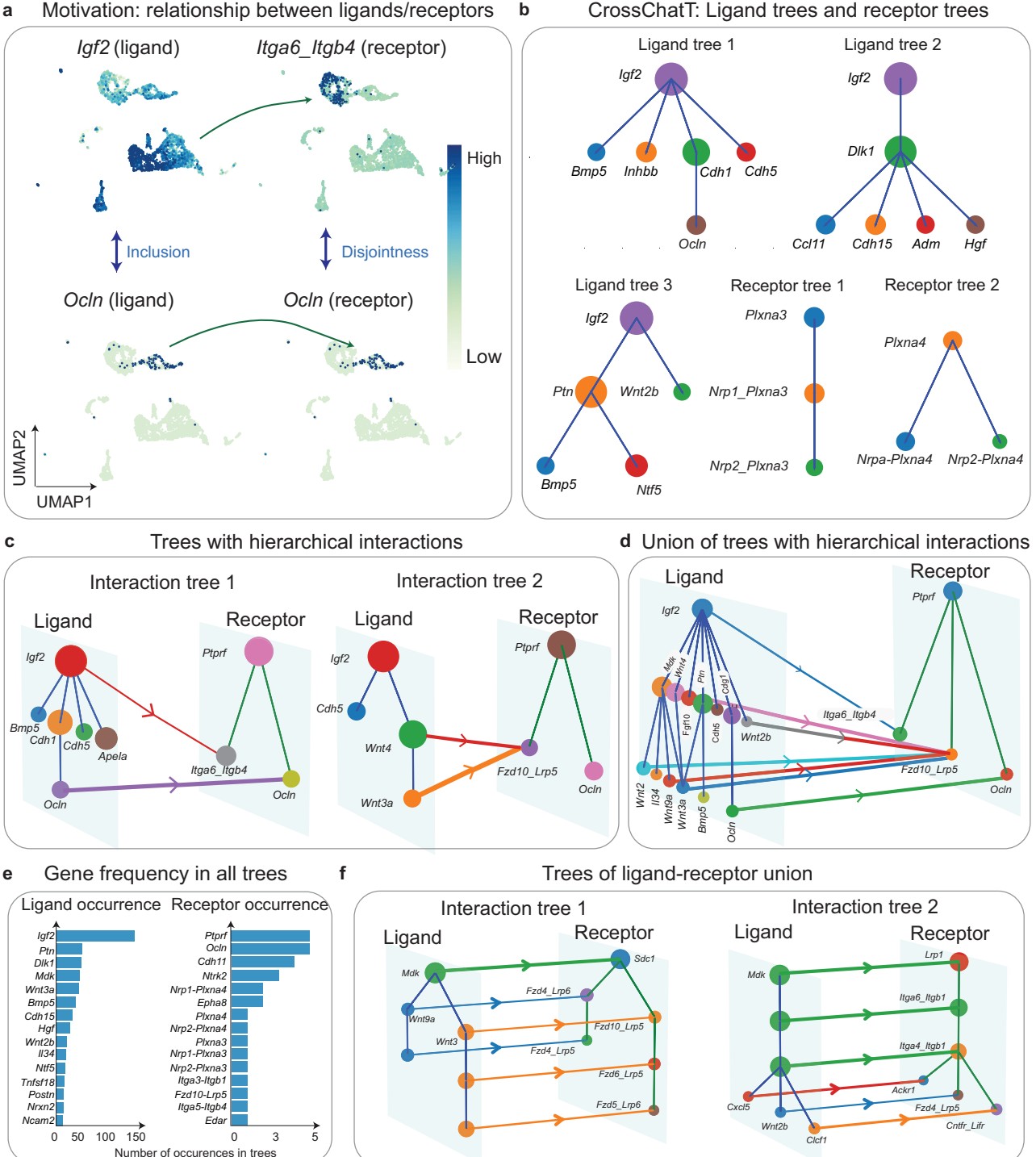

**Fig. 5 | Applications of CrossChatT to mouse skin cells during wound healing.**
**a** An example of hierarchical structures of ligand-receptor interactions. *Igf2* and *Ocln* are inclusive as ligands. *Itga6_Itgb4* and *Ocln* are disjoint as receptors.
**b** Examples of ligand tree structures and receptor tree structures detected by CrossChatT. **c** Two examples of hierarchical interactions occurring between a ligand tree and a receptor tree. **d** Hierarchical interactions between union of ligand trees and union of receptor trees. **e** Frequency of ligands/receptors occurrence across all ligand/receptor trees. **f** Examples of tree structures of ligand-receptor unions detected by CrossChat. Source data are provided as a Source Data file.

higher spatial neighborhood enrichment scores, indicating higher spatial coherence (Supplementary Fig. 4a, b). Using spatial information (see Method), CrossChatH identifies three scales of clustering (Fig. 6a). At scale 3, the clusters correspond to biologically meaningful clusters (Supplementary Fig. 5a). With the addition of spatial information, CrossChat also groups spatially adjacent spots. For example, MSCN (mid/hindbrain and spinal cord neuron) is partitioned into two

subclusters at scale 3, where subcluster 1 corresponds to neurons in the brain, and subcluster 2 corresponds to spinal cord neurons. Next, we investigated specific ligand-receptor interactions between clusters (Supplementary Fig. 5b). *Igf2-Igf2r* interaction mainly occurs within the mesoderm (Fig. 6bc). This interaction is crucial to organ development in mouse embryo, and it occurs at a large scale[50]. Interactions between synaptic adhesion molecules *Nlgn1* and *Nrxn1*, which known to be

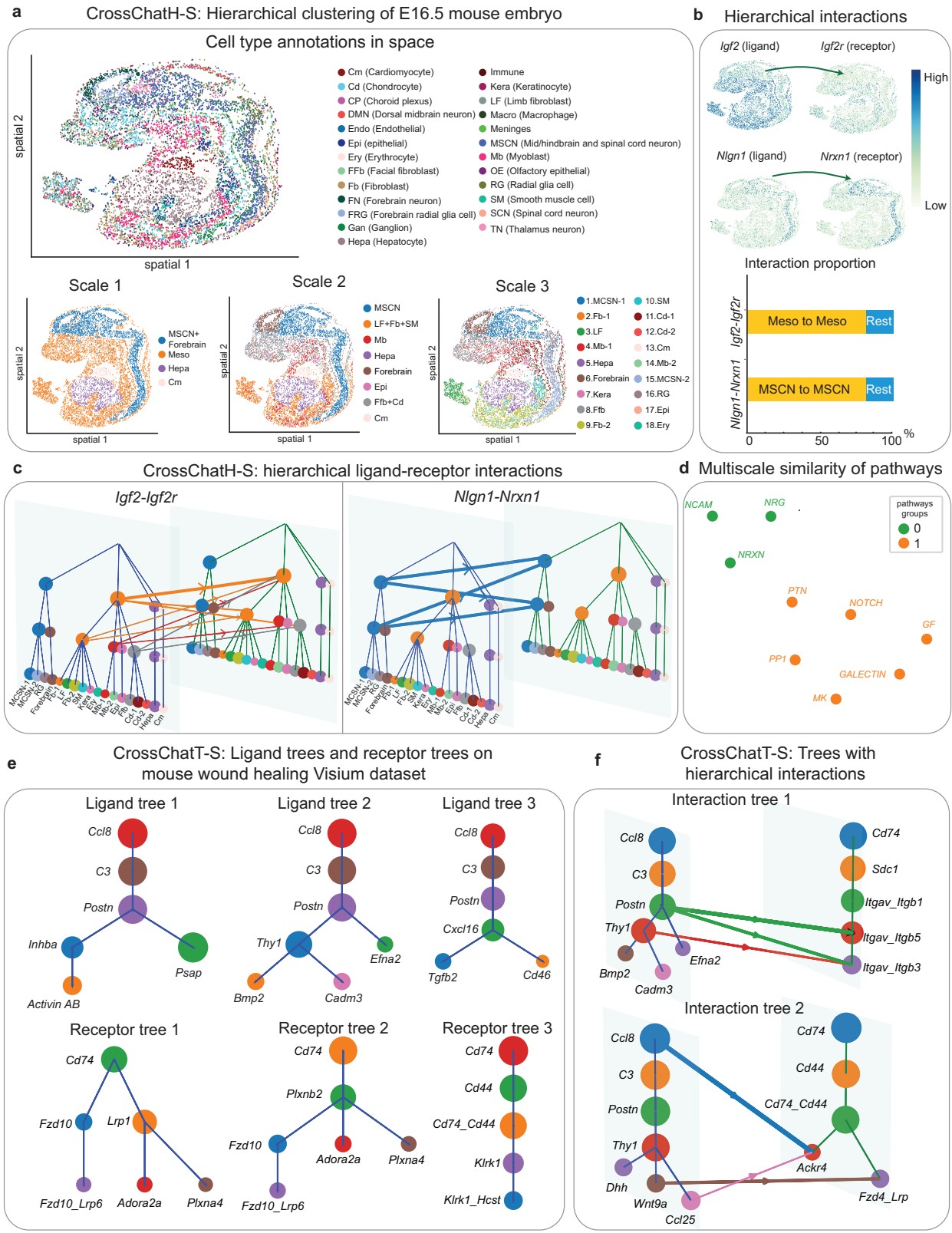

**a** CrossChatH-S: Hierarchical clustering of E16.5 mouse embryo

**b** Hierarchical interactions

**c** CrossChatH-S: hierarchical ligand-receptor interactions

**d** Multiscale similarity of pathways

**e** CrossChatT-S: Ligand trees and receptor trees on mouse wound healing Visium dataset

**f** CrossChatT-S: Trees with hierarchical interactions

crucial to synapse formation and maturation[51,52], are specifically expressed in neurons in the brain and spinal cord. Hierarchical clustering is also performed using only ligands/receptors on this dataset (Supplementary Fig. 5c–d), but hierarchical patterns of ligands/receptors were not identified. This indicates that during mouse embryo early development, where most cells have not fully matured, ligand/receptor interactions are not specialized to a particular cell group. Next, by hierarchically grouping active pathways, we detected two pathway groups (Fig. 6d). The first pathway group is associated with signaling mainly sending from brain and spinal cord neurons (MSCN), including the pathways NCAM (Neural cell adhesion molecule), NRG (Neuregulin), and NRXN (Neurexin) (Supplementary Fig. 6a). The second pathway group is associated with interactions within all other spots (Supplementary Fig. 6b). Hierarchically grouping

**Fig. 6 | Applications of CrossChatS to spatial datasets. a** Annotations and CrossChatH-S hierarchical clustering results of a Stereo-seq dataset of mouse embryo at E16.5. Spatial hierarchical clustering detects three scales of clusters, which are shown in spatial coordinates. **b** Two ligand-receptor interactions that are specific to spatial hierarchical clusters. *Ccl3* ligand is specific to Mono2, a subset of Mono, and *Ccr1* receptor is specific to Mono. *Retn* ligand is specific to Mono, and *Tlr4* receptor is specific to Mono1, another subset of Mono. Interaction proportion shows that the assigned cluster pairs contain the majority of interactions for both

*Ccl3-Ccr1* and *Retn-Tlr4* interactions. **c** Hierarchical visualization of the two ligand-receptor interactions where ligand (receptor) clusters are from CrossChatH-S clustering based on ligands (receptors) only. **d** Applying UMAP to pathways based on similarity of ligand/receptor distribution over spatial hierarchical clusters to visualize the interaction similarity in 2D. **e** Examples of ligand tree structures and receptor tree structures detected by CrossChatT-S. **f** Two examples of hierarchical interactions in space occurring between a ligand tree and a receptor tree. Source data are provided as a Source Data file.

the top 20 specific ligand-receptor pairs yields three clusters (Supplementary Fig. 6c). Group 1 includes interactions in MDK (Midkine) pathway and IGF (Insulin-like growth factor) pathway, which may both contribute to growth of the mouse embryo. Group 2 mainly contains interactions from *Dlk1* to Notch receptors, including *Notch1*, *Notch2*, and *Notch3*. *Dlk1* has been shown to inhibit Notch signaling[53]. Group 3 contains signaling between neurons, notably *Nrxn1-Nlgn3* and *Nrg3-Erbb4*.

CrossChatT-S uses spatial information to restrict the spatial distance between ligands and receptors when detect ligand/receptor trees (see Methods). Applied to a Visium dataset of wounded mouse skin sampled at post-operative day 7[54], we find many hierarchical structures within ligands and receptors during wound healing, including 124 ligand trees and 456 receptor trees (Supplementary Fig. 7a). There are 58 and 141 distinct ligands and receptors, respectively, across all ligands and receptors trees (Fig. 6e). By measuring the occurrence of each ligand and receptor (Supplementary Fig. 7b), we see that *C3*, *Postn*, *Ccl8* are most frequent across hierarchical structures in ligands, while *Cd74*, *Cd74_Cd44*, *Cd44* are highly frequent across receptor trees. The ligands *C3, Postn*, and *Ccl8* are expressed broadly by dermis-resident spots. While the *Cd74* receptor is almost expressed by all spots, expression of *Cd74_Cd44* and *Cd44* receptors is concentrated around the wounded regions (Supplementary Fig. 7ac). We then investigated interactions between ligand trees and receptor trees (Fig. 6f). In interaction tree 1, Periostin (*Postn*), a pro-fibrogenic secreted glycoprotein, interacts with its two receptors. Periostin is known to promote injury closure by facilitating the activation, differentiation, and contraction of fibroblasts[55]. *Thy1* ligand, which is expressed by a subset of *Postn*-expressing cells, affects cell adhesion and fibroblasts proliferation and migration, suggesting a more specific function of these cells during tissue repair[56]. In interaction tree 2, *Ccl8* and *Ccl25* both interact with *Ackr4*, while *Ccl25* is only expressed in a subset of *Ccl8*-expressing cells. Chemokines are major regulators of the wound-healing process[57]. *Ccl8* is known to recruit inflammatory cells including neutrophils and macrophages, which release growth factors and cytokines to improve wound healing[57]. One study found that *Ccl25* is enriched during oral wound healing process that facilitates leukocyte recruitment, which helps to protect against infection and promote wound healing[58,59]. Our finding suggests that there is a hierarchical relationship among chemokines which may facilitate specialized roles during skin wound healing.

## Comparison between CrossChat and other CCC detection methods

To show the differences between existing methods and CrossChat, we compared CrossChat with two representative CCC methods: CellChat[9] for nonspatial mouse embryonic skin scRNA-seq, and COMMOT[22] for spatial mouse embryo data. We also use NeST[60], a spatial multiscale spot detection method, to produce spatial clusters for CCC detection. By comparing CrossChatH's output for the ligand-receptor interaction, CCL3-CCR1, to CellChat, we can see that CellChat only shows that the interaction occurs between monocytes. But CrossChatH is more precise and finds that CCL3–CCR1 signaling is mainly sent from a specific subset of monocytes, Mono2 (Supplementary Fig. 8a, b). Also, by comparing CrossChatT's output with CellChat, we can see CrossChatT is able to detect the inclusion relationship between the two

interactions: *Igf2-Itga6_Itgb4*, and *Ocln-Ocln*. Specifically, the cells expressing the *Igf2* ligand also express *Ocln* ligand, and the cells expressing *Itga6_Itgb4* receptor also express *Ocln* receptor. Such hierarchical relationships among ligand-receptor interactions cannot be identified by CellChat and existing CCC methods (Supplementary Fig. 8c–e).

Furthermore, we compared the output of CrossChatH with the output of the spatial CCC method COMMOT on spatial clusters produced by NeST (Supplementary Fig. 9a–e). While NeST clustering generates multiple sparse and tiny clusters, CrossChatH detects a more general hierarchical relationship among cells, where each scale in the clustering covers all cells in the dataset (Supplementary Fig. 9c–e). By comparing CrossChatT to COMMOT where clusters are assigned by biological annotations, we again see that CrossChatT detects the structural relationship between the interactions, which cannot be detected by COMMOT. For example, the *Ccl25-Ackr4* interaction is only involved in cells which interact through *Ccl8-Ackr4* (Supplementary Fig. 9f–h).

Overall, by comparing with existing CCC detection methods, we can see that CrossChat (both CrossChatH and CrossChatT) is, in principle, different from existing methods in terms of clustering choice. CrossChat aims to detect the hierarchy of cells based on CCC relations. However existing CCC methods rely on predefined clustering and calculate the interactions between these predefined clusters. Those predefined clusters are produced by standard clustering results like Louvain community detection, which fails to consider hierarchical CCC relationships.

We also tested how different CCC calculation methods may affect the detected hierarchical structures. Using the COVID-19 scRNA-seq dataset, we recalculated CCC using NATMI[19], LIANA[61], SingleCellSignalR[14,21] (Supplementary Fig. 10a–d). Using the mouse skin wound 10x Visium data, we recalculated CCC using Giotto[21], SpatialDM[62] and spaCI[63] (Supplementary Fig. 11a–d). We can see that the hierarchical cell structures identified remain consistent across various CCC calculation methods. However, interaction strengths exhibit minor variations due to the different scoring functions employed by each method. This consistency underscores a key aspect of CrossChat's functionality: it initially identifies hierarchical structures based on only on ligand/receptor expression, independent of the CCC calculation method used. Thus, while changes in CCC calculation methods influence the inferred interaction strengths, they do not alter the underlying detected structures.

For further validation, we performed hierarchical clustering of CCC calculated using single-cell-level CCC methods, including NICHES and Scriabin on COVID-19 patients (Supplementary Fig. 12a, b). Many scattered and small clusters are detected, and they do not have hierarchical structures. This may be due to the limitations of current single-cell-level CCC methods, which do not accurately capture cell-level interactions and may introduce noise in their inference. Moreover, we attempted to explore the clustering of cell pairs using traditional hierarchical clustering and visualize the clusters using dendrograms. Specifically, we used NICHES to calculate single-cell level CCC and then performed traditional hierarchical clustering using dendrograms on cell pairs (Supplementary Fig. 12c). However, we observed one major limitation in this type of approach: although our test dataset contains only 3000 cells, there are nine million possible cell pairs, making

hierarchical clustering computationally infeasible due to both memory and complexity constraints. To gain at least a basic understanding of how traditional hierarchical clustering might work on cell pairs, we reduced the dataset to 100 cells, resulting in 10,000 cell pairs—small enough to meet computational constraints. We generated a heatmap that illustrates the overlap of sender-receiver cell groups across 20 clusters of cell pairs. However, the cell-pair structure is much less interpretable than the original cell groups, largely because different clusters of cell pairs can include identical sets of cells (simply paired differently), making it difficult to visually distinguish meaningful differences between the clusters (Supplementary Fig. 12d). For example, the Jaccard similarity of sender/receiver cells in cell pair cluster 5 and 9 is 0.98, indicating that the group of senders and receivers are indeed almost identical even though they're in two different clusters of cell-pairs. To ensure the robustness of this finding, we also performed subsampling for 20 times. We found that the average number of pairs with similarity higher than 0.5 is 11.6, indicating that many of the clusters of cell pairs have a high overlap.

## Discussion

A large number of tools are recently developed to infer cell-cell communications using either scRNA-seq or spatial transcriptomics datasets[9–24]. However, the major limitations of them are: 1. They infer CCC based on predefined cell groups, and overlook the cell structures that form with respect to ligands or receptors. 2. They overlook the existing clusters of cell states at different scales.

To overcome these limitations, we have developed CrossChat to detect hierarchical structures within CCC from scRNA-seq and spatial transcriptomics data, which infers and analyzes hierarchical CCC structures from complementary global and local perspectives. Cross-Chat can visualize CCC between hierarchical cell groups and perform downstream analysis of inferred hierarchical CCC patterns. CrossChatH detects a global hierarchical structure of cells based on gene expression across either all genes, or only across signal ligands or signal receptors, and infer hierarchical CCC patterns. CrossChatT detects hierarchical structures within ligands or receptors. Both methods can be applied to spatial transcriptomics datasets by incorporating spatial information. In general, CrossChat is particularly advantageous over existing CCC detection methods, like CellChat or CellPhoneDB in the following scenarios. First, when one wants to analyze CCC between cell groups at multiple clustering resolutions, rather than one single scale. Second, when one wants to investigate structures induced by cell-cell communication that are specific to signal ligands/receptors and independent of predefined cell type annotations.

By applying CrossChat to two nonspatial scRNA-seq datasets, PBMC cells sampled from COVID-19 patients and mouse embryonic skin at day E14.5, and two spatial transcriptomics datasets, a Stereo-seq dataset of mouse embryo at day E16.5 and a Visium dataset of mouse wounded skin[64], CrossChat is always capable of detecting biologically meaningful hierarchical structures within CCC, and improve the understanding of CCC from a structured way. As an example, Cross-ChatH recovers the hierarchical structures of PBMC cells in COVID-19 patients, and identifies specific ligand-receptor interactions within monocytes, and within subsets of monocytes, providing deeper understanding of structured functions of cells. CrossChatT detects multiple tree structures within ligands/receptors in mouse embryonic skin and provides a structural understanding of ligands/receptors in terms of their inclusion and disjointness relationship. Notably, we observed that while the growth factor, *Igf2*, expressed by many cells, the ligands *Bmp5*, *Inhbb*, *Cdh1*, *Cdh5* are expressed by cells that form mutually disjoint groups, possibly due to these cell groups performing different functions. We also observed that the adhesion gene, *Ocln*, is expressed only within a subset of *Cdh5*-expressing keratinocytes, specifically, spinous keratinocytes, suggesting a specialized role for differentiated keratinocytes.

Several directions for future studies on hierarchical patterns of CCC can be anticipated. First, while we have shown that CrossChat can be easily extended for spatial applications, there may be other ways to utilize the spatial information. For example, one may restrict CrossChatT-S analysis of hierarchical structures to a local spatial region, rather than imposing that there is a hierarchy across all cells. Furthermore, as more time-series datasets of scRNA-seq and spatial transcriptomics become available[65–67], CrossChat may be applied to gain further understanding of temporal changes of hierarchical structures within CCC. Also, with the recent availability of 3D spatio-temporal data[68], CrossChat may be used to analyze hierarchical structures within CCC in 3D, with the potential of gaining a more reliable understanding of tissues. We anticipate further studies of hierarchical structures within CCC will prosper as sequencing technologies continue to develop rapidly.

## Methods
### Data preprocessing
**COVID-19 scRNA-seq.** We analyzed a published scRNA-seq dataset generated from PBMC cells sampled from severe COVID-19 patients[64]. The data is downloaded from the NCBI GEO database at accession number GSE158055. using Scanpy[69], we removed cells expressing fewer than 1000UMI counts and cells expressing more than 25,000UMI counts, and cells with more than 10% of their total expression due to mitochondrial genes. The remaining cells were then processed through log-normalization, feature selection, and scaling. We randomly sampled 3000 cells from the preprocessed data, and used retained cell type annotations from the original dataset.

**Embryonic mouse skin scRNA-seq.** We analyzed a published mouse embryonic skin scRNA-seq dataset sampled at embryonic day E14.5[39] using Scanpy. This data is downloaded from the NCBI GEO at accession number GSM3453537. Cells expressing fewer than 2500 UMI counts or more than 50,000UMI counts were removed, as were cells with more than 10% of their total gene expression due to mitochondrial genes. We downsample 3000 cells from this dataset.

**Mouse embryo spatial Stereo-seq.** We used recently published Stereo-seq dataset of mouse embryos of embryonic day E16.5[49]. Raw data is downloaded from https://db.cngb.org/stomics/mosta/. We removed spots expressing fewer than 200 unique genes using Scanpy. The remaining spots were then processed through log-normalization, feature selection, and scaling. 10,000 spots are sampled from the data.

**Mouse wounded skin spatial Visium.** We used a recently published 10X Visium dataset of mouse-injured skin at post-operative day 7[54]. Raw data is downloaded from the NCBI GEO at accession number GSE178758. Using Scanpy, spots expressing fewer than 200 total UMI counts were removed. The dataset contains 3, 075 spots after filtering.

### Synthetic data generation
To benchmark the ability of CrossChatH to detect specific hierarchical CCC, we simulated a dataset with 1000 cells and 10,000 genes, among which 1000 are marker genes, and three clustering scales (see Fig. 3a for details). In each scale, there are two clusters, four clusters, and eight clusters respectively. This simulated dataset is based on our analysis of the characteristics of a real biological dataset of PBMC from 10X Genomics (https://cf.10xgenomics.com/samples/cell/pbmc3k/pbmc3k_filtered_gene_bc_matrices.tar.gz). The dataset contains 2700 single cells and 13,714 genes.

Step 1:

We normalize the gene expression counts of the PBMC dataset so that each cell has the same number of total counts, which is calculated as the median of the total counts across all cells. Using the cell type annotations provided by Seurat[70], we find all marker genes using

default parameters by Seurat and order marker by log fold change (based on "one *vs.* rest" comparison). We observed that, in the normalized PBMC data, the top 1,371 marker genes (10% of all genes) are among the top 20% of all genes in terms of their average expression across cells. Therefore, from 13,714 genes in the PBMC dataset, we randomly choose 1000 genes from the top 20% of genes ranked by average expression. We assign 200, 100, and 50 marker genes, respectively, to each cluster from the coarsest scale to the finest scale (Fig. 3a). There are 400, 400, and 200 marker genes in total for each scale of clusters.

Step 2:

We set the mean gene expression of our 1000chosen markers as the mean gene expression of the chosen 1000 genes from top 20% genes from the PBMC dataset. Next, at each scale, we set the fold change of marker genes as the median fold change of the top 400, 400, and 200 differentially expressed genes of the randomly chosen 1000 genes, respectively. Using the mean gene expression and median fold change of each gene, we calculated its mean gene expression both inside and outside of the cluster where the marker should be assigned.

Step 3:

As the Negative Binomial is the most common distribution used to model RNA-seq count data[71], we generated simulated gene expression counts for the 1000marker genes from a Negative Binomial distribution with the mean set as the previously calculated mean gene expression counts, and a dispersion parameter of 0.1 (default value chosen by Splatter[71]). Next, we randomly choose 9000mean expression values from the lower 80% of genes ranked by mean expression from the PBMC dataset, and simulate gene expression counts for 9000non-marker genes from a Negative Binomial distribution, using the chosen mean gene expression counts and dispersion parameter of 0.1.

## CrossChatH: hierarchical clustering

**1. KNN graph construction.** Using latent embeddings generated by PCA, we first use cosine similarity to construct a cell-cell similarity matrix. The cosine similarity is calculated either based on all genes, or ligands/receptors, based on user interest. Next, we construct a *k*-nearest neighbor (KNN) graph, setting $k = 15$. The nodes of the KNN graph are cells, and the edges represent the similarity between the cells.

**2. Random walks on graphs and Markov stability.** The KNN graph $G$ is an undirected and weighted graph. Denote the adjacency matrix of the KNN graph $G$ is as $A$. Denote the diagonal matrix, $D$, as the degree matrix of $A$, where $D_{ii} = degree(v_i)$, where $v_i$ denotes the i$^{th}$ node of the graph. We then define the random walk Laplacian of the graph $G$ as $L = I - D^{-1}A$. We then define the continuous time transition matrix of the graph $G$ as $P(t) = e^{-tL}$. The random walk transition matrix, $D^{-1}A$, has stationary distribution $\pi$, where $\pi_u$ is the limit probability of a random walk that ends up in vertex $u$ as time $t$ tends to infinity. Given communities $C_1, \ldots, C_k$ present in the the graph $G$ and an associated random walk transition matrix $D^{-1}A$ with stationary distribution $\pi$, the Markov stability[31] (MS) of community $C_i$ is defined as $MS(C_i, t) = \sum_{u,v \in V}(P(t)_{uv} - \pi_v)\delta(c_u, c_v)$, where the sum is defined over all vertices, $V$. Here, $\delta$ is the Kronecker delta defined such that $\delta(c_u, c_v) = 1$ if vertices $u$ and $v$ are in the same community and $\delta(c_u, c_v) = 0$ otherwise. Intuitively, for a given way of partitioning the graph into communities, The MS measures the overall likelihood of the random walker stays inside its community at time, $t$. If we fix the communities, this likelihood will decrease as $t$ grows, since the random walker tends to explore more other communities rather than staying in its own. For each $t$-value between zero and infinity, we use the Louvain algorithm[72] to find communities that maximizes the MS at this $t$-value. Due to the previous fact that the MS decreases with $t$ for fixed communities, we tend to have fewer (and larger) communities when

maximizing the MS for larger $t$-values, converging to the case where one community contains the whole graph.

**3. Variation of information.** The robustness of partitions of the graph, $G$, found at each time scale, $t$, is assessed using variation of information, which measures the distance between two partitions. Given two partitions of the graph $G$, $\{X_i\}_{i=1}^k$, $\{Y_j\}_{j=1}^l$, we define, $p_i = \frac{X_i}{n}$, $q_j = \frac{Y_j}{n}$, and $r_{ij} = \frac{X_i \cap Y_j}{n}$, where $n$ is the number of vertices in $G$. Then, variation of information between the two partitions is defined as $VI(\{X_i\}, \{Y_j\}) = -\sum_{i,j} r_{ij}[\log(r_{ij})/p_i + \log(r_{ij}/q_j)]$.

Before calculating the variation of information at each time $t$, we first transform $t$ to lie on the log10 scale, setting the range of $t$ to be from $10^{-1}$ to $10^4$. Next, we pick 100 evenly spaced time scales, $l_i$, ranging from $-1$ to 4, and select scales $t_i = 10^{l_i}$.

At each time scale, $t$, we run the Louvain algorithm[72] 100 times to generate 100 optimal partitions, denoted as $\{P_i\}_{i=1}^{100}$. Next, we calculate the average pairwise variation of information by $VI_g = \frac{1}{100*100}\sum_{i,j} VI(P_i, P_j)$. For each time scale $t$, we select the partition with the highest Markov stability as the representative partition, $P_t$. Then, for each pair of time scales, $t$ and $t'$, we calculate the pairwise variation of information, $VI(P_t, P_{t'})$, which forms a 100*100 matrix, $VI_P$. Intuitively, the $VI_P$ can be viewed as a graph over the lattice $\{(i,j)|1 \le i,j \le n\}$. The diagonal elements of $VI_P$ are 0, and off diagonal elements are non-negative. Thus, each diagonal entry can be viewed as the center of a basin of the graph. If the representative cluster at time scale $t$ is stable, then its pairwise $VI$ with representative partitions at neighborhood time scales will be small, thus forming a large flat basin. The optimal scales are selected by thresholding the width and depth of the basin. We then take the partitions at the selected optimal scales.

## Obtain ordered list of specific ligand-receptor pairs

First, at each scale, we use Wilcoxon rank-sum test to determine ligands and receptors that are associated significantly to a cluster, assigning the ligands and receptors to the cluster with the lowest associated *p*-value. For a ligand (receptor) multi-unit, we calculate the geometric mean of each ligand (receptor) in the complex to measure ligand (receptor) multi-unit expression. Then its *p*-value is calculated based on the ligand (receptor) multi-unit expression. Next, we only retain the ligands and receptors with associated *p*-values such that $p < 10^{-3}$. We reason that these filtered ligands and receptors are specific to its assigned clusters, due to their low *p*-values. Then, we retain ligand-receptor pairs for which both ligand and receptor are inferred marker genes of certain clusters, forming the specific ligand-receptor interaction list. We assign a specificity score each ligand-receptor pair to capture the specificity of the interaction to sender and receiver cluster, calculated by the products of *p*-values associated with the ligand and receptor, respectively. Finally, we order the specific ligand-receptor pairs their calculated scores.

## Multiscale similarity of ligand-receptor pairs and pathways

To better illustrate the relationship among multiple ligand-receptor pairs or pathways, we calculate a weighted cosine similarity between each ligand-receptor pair or between pathways of interest. For each ligand, its mean gene expression in each cluster at varying resolutions is calculated and summarized into a vector for each scale: $similarity(L_1, L_2) = \sum_{i=1}^n \frac{1}{\log(T_i)+1} cosine\_sim(u_i, v_i)$, where $n$ is number of scales, $T_i$ is the Markov time of scale $i$, $u_i$ and $v_i$ represent mean gene expression vector of $L_1, L_2$, respectively, across clusters at scale $i$. We calculate the similarity between receptors in a similar manner. We calculate the similarity between two ligand-receptor pairs as the sum of pairwise ligand similarity and receptor similarity. For two pathways, we sum the interacting ligand-receptor pairs to obtain a pathway ligands gene expression, and a pathway receptors gene expression in each cluster at varying resolutions. The weighted cosine similarity is

then calculated in a similar manner by aggregating their ligands similarity and receptors similarity. Then, we apply UMAP to the similarity matrix and use *K*-means to cluster the ligand-receptor pairs or pathways.

## CrossChatT: CCC Tree search

In order to find ligands/receptors whose supports form a hierarchical tree structure, we construct trees such that, in each tree, all pairs of ligands/receptors are either disjoint or inclusive in terms of their gene support, i.e., the cells that express these ligands/receptors. We impose that each tree has to be maximal in the sense that no other ligand/receptor can be included in this tree. We first binarize the gene expression matrix in order to find the support for each ligand/receptor (Supplementary Fig. 2). Next, we model that two ligands/receptors are connected if their relationship (with respect to their supports) is either disjoint or inclusive. Two ligands/receptors are disjoint from each other if the cells expressing them are exclusive. One ligand is a subset of another ligand if the cells expressing ligand 1 also expresses ligand 2. Thus, we form a gene relationship graph based on these relationships between ligands/receptors. In a tree of interest, all pairs of ligands/receptors are either disjoint or inclusive, thus are connected. Such a tree structure forms a complete graph, where nodes represent ligands/receptors, and every pair of nodes has an edge connecting them. In order to find largest trees of ligands/receptors in which any other ligand/receptor is either intersecting with one ligand/receptor in the tree or is disjoint with any ligand/receptor in the tree, we use Bron-Kerbosch graph search algorithm to find all maximal complete subgraphs[32]. Each complete subgraph may be one tree itself or contains multiple disjoint trees. Finally, we check each maximal complete subgraph and extract all individual trees from them (Supplementary Fig. 2).

## Incorporating spatial information

CrossChatH-S: Spatial position of both directions is first scaled into 0 to 1, and is concatenated to the PCA embeddings of cells. The concatenated PCA embeddings of cells that contain the spatial information of cells are used to calculate the similarity of cells and build the KNN graph on cells. We calculate spatial CCC using COMMOT[22].

CrossChatT-S: For any ligand or receptor, we restrict its support such that only ligands or receptors whose nearby spots express its receptors/ligands are kept in its support. Then the filtered ligands/receptors support are used to detect tree structures. Since the resolution and capture ability are different for different spatial sequencing methods, we allow users to modify their own spatial range based on their input data.

## Incorporation of CellChat and COMMOT

After CrossChatH or CrossChatT obtains the clusters at multiple resolutions, CellChat[9] (for nonspatial datasets) or COMMOT[22] (for spatial datasets) is used to calculate the interactions between the clusters.

CellChat deduces significant CCC activity from scRNA-seq data between identified cell groups. It computes an interaction score using mass action kinetics, which represents the probability of CCC. This process integrates gene expression data with existing knowledge about the interactions among signaling ligands, receptors, and their cofactors.

COMMOT infers CCC according to predefined ligand-receptor pairs by solving a global optimization problem. This method accounts for higher-order interactions among multiple ligand and receptor species. It introduces collective optimal transport to determine optimal transport plans for all species pairs simultaneously, allowing for interdependent couplings that are not possible with traditional optimal transport.

## Reporting summary

Further information on research design is available in the Nature Portfolio Reporting Summary linked to this article.

## Data availability

The datasets analyzed in this study are available from the Gene Expression Omnibus (GEO) repository under the following accession numbers: GSE158055 (COVID-19), GSM3453537 (mouse skin), and GSE178758 (mouse wounded skin). Mouse embryo dataset can be found in the following link: https://db.cngb.org/stomics/mosta/. Source data are provided with this paper.

## Code availability

The CrossChat package[73] is implemented in Python and is available on the GitHub repository https://github.com/Xinyiw28/CrossChat. The tool is deposited in PyPI for installation: https://test.pypi.org/project/crosschat/0.0.1/. A Read the Docs website can be found here: https://crosschat.readthedocs.io/en/latest/README.html. It is also deposited at Zenodo: https://zenodo.org/records/13984908.

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

## Acknowledgements

This project was supported by grants from National Institutes of Health grants R01AR079150, R01DE030565, and R01GM152494, National Science Foundation grants DMS1763272 (Q.N.) and CBET2134916, and The Simons Foundation (594598 to Q.N.).

## Author contributions

X.W., A.A.A., and Q.N. conceived the project. A.A.A. and Q.N. supervised the research. X.W. developed and implemented the computational approach. X.W., A.A.A., and Q.N. wrote the manuscript. All authors read and approved the final manuscript.

## Competing interests

The authors declare no competing interests.
