## [Transparent Peer Review file · Nature Communications]

Detecting global and local hierarchical structures in cell-cell communication using CrossChat

Corresponding Author: Dr Qing Nie

Version 0:

Reviewer comments:

Reviewer #1

(Remarks to the Author)

Wang et al. developed a new computational framework, called CrossChat, for identifying hierarchical cell-cell communication (CCC) patterns using single-cell transcriptomics data or spatial transcriptomics data. The concept of global and local CCC structures proposed in this paper is novel and interesting. The authors applied CrossChat to two scRNA-seq dataset (PBMC cells in COVID-19 patients and mouse embryonic skin) and two spatial transcriptomics datasets (mouse embryo and mouse wounded skin) and showed that structured and biological meaningful CCC patterns across different scales can be revealed. Overall, I believe CrossChat is a useful tool for analysis of hierarchical CCC patterns and will facilitate the deep understanding of CCC mechanisms within tissue microenvironments.

Major comments:

1. Although I agree there exist hierarchical CCC patterns within the tissue, it seems the example (i.e., interaction between T follicular helper cells and B cells) for explanation of the hierarchical structures in CCC provided in the Introduction section is not appropriate. The hierarchical CCC should not be explained based on different interpretations on a specific ligand-receptor (L-R) interaction. Maybe it's better to show some examples that T cells and B cells interact via some specific L-R interactions and a subset of T cells simultaneously interact with B cells (or its subset) via some other specific L-R interactions.
2. The robustness of the method should be evaluated with respect to the K value in KNN graph construction. What is the rationale for setting $K=15$? Will the results substantially change when the value of K is changed?
3. CrossChatH was demonstrated to have the ability to detect cluster marker genes across scales, but how is this related to its performance in hierarchical CCC detection? More clarifications are needed.
4. When applying CrossChatH to COVID-19 patients, the authors found several hierarchical CCC patterns related to monocytes and its subsets. And then several literatures were used for supporting their findings. But those literatures only demonstrated the important roles of those ligands or receptors in COVID-19 but not explicitly related with hierarchical CCC, which makes the evidence not solid.
5. I totally understand the performance evaluation is the most challenging work in CCC studies. However, quantitative evaluation of CrossChatT would significantly enhance the manuscript. And I also suggest to perform method comparisons in order to demonstrate the superiority of CrossChat (especially for CrossChatT) over existing CCC methods in identification of hierarchical CCC patterns, even if existing methods were not originally designed for hierarchical CCC detection. For instance, is CrossChat able to detect biological meaningful L-R interactions between different scales, which cannot be identified by existing CCC methods?

Minor comments:

1. There is no line number throughout the manuscript, which caused inconvenience to directly point out issues in

expression.

2. On the second line under the “Overview of CrossChat” subsection, using “clustering cells” seems better than “clustering clusters”.
3. It seems the high-resolution figures are missing.
4. In Figure 1e, “Ligand 1” should be consistently used.
5. Under the subsection “CrossChatH identifies ... COVID-19 patients”, the sentence “we assigned each cluster to a cell type” was repeated.

(Remarks on code availability)

Reviewer #2

(Remarks to the Author)

Wang, Almet and Nie present CrossChat, a new computational method to infer and analyze hierarchical structures in cell-cell communication (CCC) from single-cell and spatial transcriptomics data. The key novelty is that CrossChat detects CCC patterns at multiple resolutions of cell cluster annotation, rather than just predefined cell types as done by existing methods. It does this through two complementary approaches:

CrossChatH, which detects a global hierarchy of cell clusters and CCC between them using multiscale community detection.

CrossChatT, which detects local hierarchical structures among ligands, receptors, and their interactions using a tree-based approach.

The method is tested on simulated and real datasets, providing extensions on spatial data as well. While the method shows potential for extracting interesting information that has been previously missed by other tools, the authors need to address some major concerns to convince readers of the value and significance of their method among the many available tools for studying cell-cell communication. Specifically, the authors should improve the explanation of the method and include benchmarking. I provide comments and suggestions to address this.

Major comments:

1. The importance of finding hierarchical structures of CCC is unclear. Improvements regarding this concept should be made in different parts of the manuscript.

- a. The authors mention the existence of hierarchical structures early in the abstract, but it is unclear what they mean by these structures.

- b. Similarly, the authors should clarify in the Introduction how the global and local hierarchy concepts differ and complement each other. Consider including an intuitive toy example, as the example of T cells and B cells in the first paragraph is not clear enough.

- c. The authors should explain their rationale for focusing on finding hierarchical structures using multiple distinct resolutions simultaneously (or subclusters defined by their CrossChatH method) rather than inferring CCC for high-resolution clusters. This explanation should be provided either in the Introduction or Discussion.

2. Lack of single-cell resolution: While CrossChat evaluates CCC across multiple resolutions of clustering, including cell subtypes, a major limitation is that it does not provide CCC inference at the single-cell level like some other tools (e.g., NICHES, <https://doi.org/10.1093/bioinformatics/btac775>, and Scriabin, <https://doi.org/10.1038/s41587-023-01782-z>). The biological relevance of finding hierarchical structures of CCC by varying the clustering resolution vs. from the CCC of individual cells should be discussed further.

Suggestion: The authors should provide a benchmarking or comparison of methods where a baseline is inferring CCC with single-cell resolution using NICHES and/or Scriabin. Then, perform hierarchical grouping of cell pairs from their CCC and identify key communication patterns. With these results, the authors should compare what biological insights could be obtained in terms of CCC hierarchy with each approach. Additionally, they should discuss the trade-offs and justify the focus on multi-resolution cluster-level analysis done by CrossChat.

3. Sensitivity of hierarchical clusters: The authors should evaluate the robustness of the detected cluster hierarchies to noise, sparsity, and normalization of the data.

Suggestion: Include sensitivity analyses that evaluate these factors and report the robustness of the main biological findings to the assessed variations. Furthermore, the authors should evaluate alternative ways of finding cluster hierarchies (by using other tools designed for this purpose, <https://doi.org/10.1261/rna.078965.121>).

4. More details should be provided about the methodology. For example:

- a. When building the cell-cell similarity graph based on cosine similarity from the principal components, did the authors use the whole gene expression matrix or just ligands and/or receptors?
- b. The use of CellChat and COMMOT was mentioned for inferring CCC in single-cell and spatial data, but it is not clear how and when these scores were computed and passed to their approach.
- c. The method section about CCC tree search should be better explained, specifically regarding what is a disjoint or inclusive relationship and how this is included and used by the algorithm. Consider adding a supplementary figure representing how these relationships are found from the binary values and how the trees are built (explicitly showing connections for a toy example).

5. More details about the selection of parameters and analysis decisions should be provided. Furthermore, a benchmarking of different strategies and parameters should be provided. For example:

- a. The authors should clarify the rationale for using the geometric mean for aggregating p-values of all genes in the ligand or receptor complex. Multiple approaches exist (e.g., <https://www.pnas.org/doi/10.1073/pnas.0406811102>, <https://doi.org/10.1371/journal.pone.0125081>, <https://www.nature.com/articles/s41598-021-86465-y>), so it would be important to indicate why they chose the geometric mean over other approaches. Consider providing a comparison using a couple of different methods to show how they could affect the results.
- b. The authors should compare the use of tools other than CellChat and COMMOT for inferring CCC from single-cell and spatial data and evaluate how the hierarchical structures change. See alternative methods here: <https://doi.org/10.1038/s41576-023-00685-8>.
- c. What parameters were used for binarizing the gene expression matrix in the CCC tree search? Different ways of binarizing these values should be provided, showing how this could affect the detection of local hierarchy.

6. Implement an automatic approach for finding important structures of LR pairs: While the authors highlight important pairs of clusters with distinct hierarchies and key LR interactions, it is unclear how they selected them. It seems that a certain level of expertise is needed to fully use the tool's capabilities.

Suggestion: The authors should implement an unsupervised way of selecting important hierarchical structures that could be pertinent to a dataset of interest. Alternatively, they should provide guidance on what to focus on for selecting important structures (maybe include a brief section for this).

7. For the sake of reproducibility and tool usage, the authors should improve the documentation of CrossChat by including, for example, a readthedocs website. In addition, they should deposit the tool in PyPI or conda to facilitate its installation. Finally, they should incorporate changes related to my other comments into their tutorial, making the tutorials as comprehensive as possible.

Minor comments:

1. The detail on the PBMC dataset simulation (page 6) can be moved to Methods or Supplementary Information to improve readability. Just convey the key points in the Results.
2. Provide more intuition on the Markov stability metric and explain the resolution parameter t in the CrossChatH Methods.
3. The word "multiscale" sounds a bit ambiguous and sometimes hard to follow. From my point of view, it could be better to use something like "varying resolution" or other alternatives instead.
4. Does the method allow the inclusion of a predefined hierarchy? For example, using the hierarchy of cells seen during development along time (progenitor-precursor-differentiated cell relationship). This should be discussed, and maybe an example could be provided in the tutorials.

(Remarks on code availability)

For the sake of reproducibility and tool usage, the authors should improve the documentation of CrossChat by including, for example, a readthedocs website. In addition, they should deposit the tool in PyPI or conda to facilitate its installation. Finally, they should incorporate changes related to my other comments into their tutorial, making the tutorials as comprehensive as possible.

Reviewer #3

(Remarks to the Author)

This topic is great. To the best of my knowledge, it is indeed the first method to detect hierarchical structures within CCC.

The author also demonstrated through the explanation in the "Introduction" section and the analysis of three real data using CrossChat in the "Results" section that hierarchical structures within CCC can indeed bring effective biological discoveries. The authors also used simulation data for validation, while adapting to single-cell and spatial transcriptome data, giving CrossChat a wider range of application scenarios. CrossChat does have a good appeal to biologists and can also advance algorithm research in the field of intercellular communication. I think that this article can be considered for acceptance after appropriate modifications.

Major points:

1. The author only used simulation data to verify the function of detecting specific CCC of CrossChatH, which is only a small part of the CrossChat function. Can the author provide a validation for functions from "CrossChatT"?
2. A more detailed description is needed on how CrossChat uses spatial information, especially regarding whether CrossChat H-S can effectively utilize spatial information by "concatenating to the PCA embeddings of cells". If there is a precedent for this approach, please include appropriate references. Alternatively, can the author compare the results with and without spatial information based on spatial transcriptomic data to verify that CrossChat effectively utilizes spatial information?
3. For CrossChatT-S, the author uses the following settings: "For any ligand or receptor, we restrict its support such that only ligands or receptors whose nearby spots express its receptors/ligands are kept in its support". However, there are significant differences in resolution and capture ability among different spatial sequencing methods. For spatial transcriptomic data with very sparse gene expression, only considering adjacent spots will result in only a small number of LR pairs being detected. Can the author set a parameter so that users can choose to include CCC within which spatial distance range based on their own data?

Minor points:

1. The caption of Figure 2 includes a, b, and c, but only a and b are present in the image. Maybe both a and b in the caption correspond to Figure 2a.
2. In the leftmost subgraph of Figure 2b, the annotation for the blue subgroup is missing and should match the red, yellow, and blue subgroups.
3. Is there a corresponding relationship between (scale 2)-(cluster1-8) in Figure 1a and each leaf node (C4-C11) in Figure 1d? If so, maybe the authors should consider naming each leaf node in Figure 1d as SC1-SC8.
4. Consider moving some of the content from the section "Validation of CrossChat using simulated dataset and COVID-19 dataset" into the Methods section and re-organizing it in a step-by-step manner.
5. A preprocessed toy data that meets the CrossChat input requirements should be provided in the tutorial of Github, or a toy data download link should be provided.
6. The name of each node, at least the nodes connected by the edges of CCC, should be provided in the output image of the "ccH_obj.Draw_CCC_LR" function.

(Remarks on code availability)

Version 1:

Reviewer comments:

Reviewer #1

(Remarks to the Author)

Thank you to the authors for their comprehensive responses to my previous concerns. All of those concerns have been adequately addressed in the revised manuscript. I have only two minor suggestions that could be incorporated by minor text editing.

Minor comments:

1. Several traditional CCC inference methods have been widely used in single-cell and spatial transcriptomics studies, such as CellChat and CellphoneDB. Could the authors provide a short guideline in the Discussion section for selecting their proposed CrossChat method over traditional approaches? Under what circumstances might CrossChat be particularly advantageous?
2. Supplementary Fig. 1 referenced on Line 236, 248, 275, 285 and 288 should be Supplementary Fig. 3.

(Remarks on code availability)

Reviewer #2

(Remarks to the Author)

I appreciate the authors' efforts in addressing the previous review comments and improving their manuscript. The changes have significantly enhanced the quality of the work. However, there are a few important points that should be addressed before publication:

1. Many analyses performed in response to reviewer comments should be incorporated into the manuscript since they could be informative to readers wondering about similar ideas. These should be presented as supplementary figures or notes and referenced in the main text. Specifically:

- a) Figures responding to reviewer #2's point #2 (single-cell tools + hierarchical clustering on CCC scores)
- b) Figures for reviewer #2's point #5b (including other CCC tools for predicting communication used as input of CrossChat)
- c) Figures for reviewer #2's point #5c (using different thresholds for binarizing gene expression)
- d) Any other relevant analyses not currently included

Please integrate these analyses and discuss or mention them, even if briefly, to provide readers with a comprehensive understanding of the methodology and its robustness.

2. While the authors have addressed the use of single-cell tools and clustering (reviewer #2's point #2), the intention of my original comment was slightly different. My original thought was about using this alternative approach as a baseline. This approach would through using tools like NICHES or Scriabin to infer a CCC matrix of single-cell sender-receiver pairs (rows/columns) by all LR pairs (columns/rows). Then, perform traditional hierarchical clustering, instead of CrossChat approach, using the whole vector per cell pair (i.e., all scores of LR pairs in a pair of cells). You can run for example the hierarchical clustering employed to generate dendrograms in scipy or seaborn clustermaps. Finally, compare the resulting substructures with those obtained from CrossChat.

Please conduct this analysis and discuss how the results compare to CrossChat's.

3. For reviewer #2's point #5b: The current comparison with NATMI and Giotto is a good start, but not conclusive to say that CrossChat's result do not change across CCC tools given the small number of tools employed here. I recommend including at least a couple of extra tool in each case, covering distinct strategies to infer CCC:

- a) For single-cell data: LIANA (using aggregated rank across multiple methods) and SingleCellSignalR (with regularized score)
- b) For spatial data: SpatialDM (based on auto-correlation Moran's I), DeepLinc, and spaCI (both based on deep learning but with different loss functions)

Please expand the comparison to include these additional tools and discuss the implications of the results.

4. While the ReadtheDocs documentation provides useful tutorials, it lacks detailed API documentation. Please add comprehensive API documentation, including docstrings and information about input/output parameters for each function in the codebase. This will greatly assist users in understanding the tool's functionality at each step.

(Remarks on code availability)

Reviewer #3

(Remarks to the Author)

My concerns have been addressed in the revisions.

(Remarks on code availability)

Version 2:

Reviewer comments:

Reviewer #2

(Remarks to the Author)

I congratulate the authors for their work. I am now satisfied with the changes made, and they have addressed all of my comments. I hope this tool will be useful to gain new insights in cell-cell communication that other methods were missing.

(Remarks on code availability)

REVIEWER COMMENTS

Reviewer #1 (Remarks to the Author):

Wang et al. developed a new computational framework, called CrossChat, for identifying hierarchical cell-cell communication (CCC) patterns using single-cell transcriptomics data or spatial transcriptomics data. The concept of global and local CCC structures proposed in this paper is novel and interesting. The authors applied CrossChat to two scRNA-seq dataset (PBMC cells in COVID-19 patients and mouse embryonic skin) and two spatial transcriptomics datasets (mouse embryo and mouse wounded skin) and showed that structured and biological meaningful CCC patterns across different scales can be revealed. Overall, I believe CrossChat is a useful tool for analysis of hierarchical CCC patterns and will facilitate the deep understanding of CCC mechanisms within tissue microenvironments.

Response: Thank you for your appreciation of CrossChat and for the insightful comments. Substantial improvement has been made in the revision, and multiple changes are highlighted in red throughout our revised manuscript. Below are our detailed responses.

Major comments:

1. Although I agree there exist hierarchical CCC patterns within the tissue, it seems the example (i.e., interaction between T follicular helper cells and B cells) for explanation of the hierarchical structures in CCC provided in the Introduction section is not appropriate. The hierarchical CCC should not be explained based on different interpretations on a specific ligand-receptor (L-R) interaction. Maybe it's better to show some examples that T cells and B cells interact via some specific L-R interactions and a subset of T cells simultaneously interact with B cells (or its subset) via some other specific L-R interactions.

Response: Thank you for your comment. We agree that it is better to explain hierarchical ligand-receptor interactions using a specific interaction pair. In the updated Introduction section, we used an example of two ligand-receptor interaction pairs, where one involves

CD4+ T cells to B cells and another involves a subset of CD4+ T cells and B cells, to better explain the hierarchical structures arising from CCC.

2. The robustness of the method should be evaluated with respect to the K value in KNN graph construction. What is the rationale for setting K=15? Will the results substantially change when the value of K is changed?

Response: Thank you for your comment. Theoretically, when K is larger, graph construction becomes more computationally intensive. When K is smaller, the graph will have lower connectivity. For all of our experiments, K=15 worked well when determining multiscale clusters. However, we also allow users to choose their desired value for K when constructing the KNN graph for the dataset in question. To evaluate the robustness of multiscale clustering with respect to K, the number of nearest neighbors, during KNN graph construction, we ran the method on a PBMC dataset for the following K values: 5, 10, 15, 20, 25, and 50. We calculate the multiscale similarity between clustering of different K values, and observed that the score between each multiscale clustering is almost always greater than 0.9, indicating a high robustness (See Robustness of CrossChatH in supplementary materials).

3. CrossChatH was demonstrated to have the ability to detect cluster marker genes across scales, but how is this related to its performance in hierarchical CCC detection? More clarifications are needed.

Response: Thank you for the comment. In our updated manuscript, we have added explanations about the relation between hierarchical marker gene detection and hierarchical CCC detection in the Results section, "Validation of CrossChat using simulated dataset and COVID-19 dataset". Specifically, when all genes are used for clustering, hierarchical clusters are first detected, and CCC is then calculated based on the clusters. This way, CrossChatH does not aim to "directly" detect hierarchical CCC, but rather, to first find hierarchical cell clusters, then infer CCC between these hierarchical clusters. By using only ligands/receptors for clustering, CrossChatH can detect hierarchical clusters that have statistically significant CCC patterns. The detected markers

for each cluster in the hierarchy of cells are ligands/receptors, so it can be interpreted as a hierarchical CCC.

Furthermore, the clusters detected by CrossChatH rely significantly on the input genes used, which influence the detected hierarchical CCC. For example, we use CrossChatH on a mouse embryonic skin data (PMID: 33597522), using three different input: all genes, only ligands, and only receptors. From the figure below, we can see that the detected hierarchical clusters slightly vary from each other, and the detected hierarchical CCC (specific to the hierarchical clusters) also vary correspondingly.

b Top 20 ligand-receptor interactions detected by CrossChatH

Input: all genes

	Ligand	Receptor	Specificity
1	Ptn	Ncl	0
2	Angpt4	Cdh11	0
3	Dlk1	Notch4	0
4	Dlk1	Notch3	0
5	Dlk1	Notch2	0
6	Dlk1	Notch1	0
7	Ptn	Sdc4	0
8	Igf2	Igf2r	0
9	Mdk	Ncl	0
10	Pdgfa	Pdgfra	0
11	Wnt6	Fzd10 Irp6	0
12	Pthlh	Pth1r	0
13	Pdgfa	Pdgfrb	0
14	Cdh1	Cdh11	0
15	Wnt5a	Fzd10	0
16	Ptn	Sdc1	0
17	Itga4 Itgb1	Vcam1	0
18	Pdgfc	Pdgfra	1.86E-318
19	Itga9 Itgb1	Vcam1	2.77E-307
20	Wnt5a	Fzd2	2.04E-305

Input: ligands/receptors

	Ligand	Receptor	Specificity
1	Dlk1	Notch4	0
2	Angpt4	Cdh1	0
3	Cdh1	Cdh1	0
4	Pdgfc	Pdgfra	0
5	Dlk1	Notch3	0
6	Dlk1	Notch2	0
7	Dlk1	Notch1	0
8	Itga4 Itgb7	Vcam1	0
9	Igf2	Igf2r	3e-323
10	Mdk	Ncl	8.62e-311
11	Wnt5a	Fzd2 Irp6	7.28e-304
12	Pdgfc	Pdgfra	5.48E-290
13	Itga4 Itgb7	Vcam1	6.09E-290
14	Lrrc4b	Ptpfr	1.96E-285
15	Ncam	Fgfr1	4.45E-285
16	Pthlh	Pth1r	9.92E-280
17	Itga4 Itgb7	Vcam1	2.22E-279
18	Wnt5a	Fzd10	2.54E-274
19	Igf2	Itga6 Itgb4	2.09E-270
20	Wnt5a	Fzd2	8.19E-260

4. When applying CrossChatH to COVID-19 patients, the authors found several hierarchical CCC patterns related to monocytes and its subsets. And then several literatures were used for supporting their findings. But those literatures only demonstrated the important roles of those ligands or receptors in COVID-19 but not explicitly related with hierarchical CCC, which makes the evidence not solid.

Response: Thank you very much for this comment. To the best of our knowledge, there is no existing literature that discusses the relationship between CCL3-CCR1 and RETN-TLR4 interactions. This paragraph is focused on new biological findings of the existence of hierarchical CCC, and their potential implications to biology. While not much discussion has been made on hierarchical structures in CCC in literature, we hope they will become a larger part of CCC analysis in the future.

5. I totally understand the performance evaluation is the most challenging work in CCC studies. However, quantitative evaluation of CrossChatT would significantly enhance the manuscript. And I also suggest to perform method comparisons in order to demonstrate the superiority of CrossChat (especially for CrossChatT) over existing CCC methods in identification of hierarchical CCC patterns, even if existing methods were not originally designed for hierarchical CCC detection. For instance, is CrossChat able to detect biological meaningful L-R interactions between different scales, which cannot be identified by existing CCC methods?

Response: Thank you very much for your comments on improving our paper by including better evaluations of our methods.

Following your suggestions, we first performed experimental validation of the algorithm in our setting. Specifically, we generate 1,000 cells with 100 genes in each cell. We distributed the gene expression so that there are at least 5 trees as shown in the figure, and there are a total of 15 genes involved in these 5 trees (Fig. 3e). For each of the remaining 85 genes, we randomly chose 500 cells to express them. We ran the experiment for 1000 times. CrossChatT is able to detect all 5 trees we generated in each trial (Fig. 3e).

To experimentally show the differences between existing methods and CrossChat, we compared CrossChat with two representative CCC methods: CellChat (PMID: 33597522) for nonspatial mouse embryonic skin data, and COMMOT (PMID: 36690742) for spatial

mouse embryo data. We also use NeST (PMID: 37848426), a spatial multiscale spot detection method, to produce spatial clusters for CCC detection. By comparing CrossChatH's output for CCL3–CCR1 interactions to CellChat, we can see that CellChat only shows that the interaction is between monocytes. But CrossChatH is more precise and finds that CCL3-CCR1 signaling is mainly sent from a subset of monocytes, Mono2 (Supplementary Fig. 8a). Also, while CellChat detects that RETN-TLR4 interaction is from monocytes to monocytes, CrossChatH can precisely detect that the receptor is indeed expressed by only a subset of monocytes (Supplementary Fig. 8b). By comparing CrossChatT's output with CellChat, we can see CrossChatT is able to detect the inclusion relationship between the two interactions: *Igf2* to *Itga6_Itgb4*, and *Ocln* to *Ocln*. Specifically, the cells expressing the *Igf2* ligand also express *Ocln* ligand, and the cells expressing *Itga6_Itgb4* receptor also express *Ocln* receptor. Such hierarchical relationship among ligand-receptor interactions cannot be identified by CellChat and existing CCC methods (Supplementary Fig. 8c-e).

Furthermore, we compared the output of CrossChatH with the output of the spatial CCC method COMMOT, on spatial clusters produced by NeST (Supplementary Fig. 9a-e). While NeST clustering generates multiple sparse and tiny clusters, CrossChatH detects a more general hierarchical relationship among cells, where each scale in the clustering covers all cells in the dataset (Supplementary Fig. 9c-e). By comparing CrossChatT to COMMOT where clusters are assigned by biological annotations, we again see that CrossChatT detects the structural relationship between the interactions, which cannot be detected by COMMOT. For example, the *Ccl25-Ackr4* interaction is only involved in cells which interact through *Ccl8-Ackr4* (Supplementary Fig. 9f-h).

Overall, by comparing with existing CCC detection methods, we can see that CrossChat (both CrossChatH and CrossChatT) is, in principle, different from existing methods in terms of the choice of clusters. CrossChat aims to **detect the hierarchy** among cells based on their CCC-based relationships. But existing CCC methods rely on a predefined clustering result, and aims to **calculate the interactions** between the **predefined**

clusters (cell types). Those predefined clusters are produced by standard clustering workflows that, for example, use the Louvain community detection method, which fail to account for hierarchical CCC.

In our updated manuscript, we have added the results on quantitative evaluation of CrossChatT, and comparison of CrossChat with existing methods to show CrossChat is superior.

Minor comments:

1. *There is no line number throughout the manuscript, which caused inconvenience to directly point out issues in expression.*

Response: Sorry for the inconvenience. In our updated manuscript, we have added line numbers.

2. *On the second line under the “Overview of CrossChat” subsection, using “clustering cells” seems better than “clustering clusters”.*

Response: Thank you for the suggestion. We have changed accordingly in our revised manuscript.

3. *It seems the high-resolution figures are missing.*

Response: Thank you for the comment. We have updated our figures to be high resolution.

4. *In Figure 1e, “Ligand 1” should be consistently used.*

Response: Thank you for the comment. We have modified accordingly by changing “Gene1” to “Ligand 1” for consistent annotation.

5. *Under the subsection “CrossChatH identifies ... COVID-19 patients”, the sentence “we assigned each cluster to a cell type” was repeated.*

Response: Thank you for the comment. We have deleted the repeated sentence “we assigned each cluster to a cell type”.

Reviewer #2 (Remarks to the Author):

Wang, Almet and Nie present CrossChat, a new computational method to infer and analyze hierarchical structures in cell-cell communication (CCC) from single-cell and spatial transcriptomics data. The key novelty is that CrossChat detects CCC patterns at multiple resolutions of cell cluster annotation, rather than just predefined cell types as done by existing methods. It does this through two complementary approaches:

CrossChatH, which detects a global hierarchy of cell clusters and CCC between them using multiscale community detection. CrossChatT, which detects local hierarchical structures among ligands, receptors, and their interactions using a tree-based approach.

The method is tested on simulated and real datasets, providing extensions on spatial data as well. While the method shows potential for extracting interesting information that has been previously missed by other tools, the authors need to address some major concerns to convince readers of the value and significance of their method among the many available tools for studying cell-cell communication. Specifically, the authors should improve the explanation of the method and include benchmarking. I provide comments and suggestions to address this.

Response: Thank you for your insightful comments and suggestions on improving our manuscript. We have made several revisions to our manuscript and believe that they have improved the work substantially. Multiple changes are highlighted in red throughout our revised manuscript. Below are our detailed responses to your comments and suggestions.

Major comments:

1. *The importance of finding hierarchical structures of CCC is unclear. Improvements regarding this concept should be made in different parts of the manuscript.*

a. *The authors mention the existence of hierarchical structures early in the abstract, but it is unclear what they mean by these structures.*

Response: This is a good point. Following your suggestion, in the abstract, we have added clarifications on the existence of hierarchical structures in CCC. Specifically, we explained that the hierarchical structures of CCC form when the interacting cell clusters exist at multiple resolutions. From a global perspective, CCC may exist between cell groups at varying scales, instead of one level of cell clustering. From a local perspective, a group of cells, that secretes one ligand A, may contain two mutually disjoint subgroups of cells that secrete ligand B and ligand C, respectively.

b. *Similarly, the authors should clarify in the Introduction how the global and local hierarchy concepts differ and complement each other. Consider including an intuitive toy example, as the example of T cells and B cells in the first paragraph is not clear enough.*

Response: Thank you for your comment. In our updated introduction section in manuscript, to better explain the global hierarchical structures in CCC, we gave an illustration involving two ligand-receptor interactions, where one took place between CD4⁺ T cells to B cells, and the other involved a subset of CD4⁺ T cells to B cells. Furthermore, we explained that there are genes expressed during a specific phase of a cell cycle among many different cell types, and they perform important biological functions. For example, senescent cells induce significant paracrine effects on other cells and the tissue microenvironment via the senescence-associated secretory phenotype (PMID: 38654098). Therefore, considering cells that express a certain ligand/receptor, instead of cells that belong to a specific cell type (global hierarchy), is more biologically relevant given the existence of these genes. Thus, we need a local hierarchy in CCC that

is determined completely by the cells where ligands/receptors involved in the interaction are expressed.

c. The authors should explain their rationale for focusing on finding hierarchical structures using multiple distinct resolutions simultaneously (or subclusters defined by their CrossChatH method) rather than inferring CCC for high-resolution clusters. This explanation should be provided either in the Introduction or Discussion.

Response: Thank you for your comment. In the first paragraph of our revised Introduction section, we have included more explanations on the necessity of a hierarchy. In the example of a hierarchy in CCC related to cell types at varying scales, if we only consider one scale of clustering, where T cells are a cluster, the two interactions sending from T follicular helper cells, and CD4+ T cells respectively, will both be assigned as interactions from T cells. A coarse scale of clustering will lose precision in ascribing the interactions to cell types. However, focusing solely on the finest scale of clusters might cause us to miss the broader functions of T cells and B cells, which collaborate to execute specific biological functions. Therefore, it's essential to consider the global hierarchy of clusters rather than just a single "scale" of clusters.

2. Lack of single-cell resolution: While CrossChat evaluates CCC across multiple resolutions of clustering, including cell subtypes, a major limitation is that it does not provide CCC inference at the single-cell level like some other tools (e.g., NICHES and Scriabin). The biological relevance of finding hierarchical structures of CCC by varying the clustering resolution vs. from the CCC of individual cells should be discussed further. Suggestion: The authors should provide a benchmarking or comparison of methods where a baseline is inferring CCC with single-cell resolution using NICHES and/or Scriabin. Then, perform hierarchical grouping of cell pairs from their CCC and identify key communication patterns. With these results, the authors should compare what biological insights could be obtained in terms of CCC hierarchy with each approach. Additionally, they should discuss the trade-offs and justify the focus on multi-resolution cluster-level analysis done by CrossChat.

Response: Thank you for your suggestion. We agree this is an interesting approach to study hierarchical structures in CCC. Following your suggestion, we have calculated CCC using NICHES (PMID: 36458905) and Scriabin (PMID: 35169794) on Peripheral Blood Mononuclear cells of COVID-19 patients (PMID: 33657410), and performed hierarchical clustering on those cells using CrossChatH. We found that many scattered and small clusters are detected, and they do not have hierarchical structures. This may be due to the limitations of current single-cell level CCC methods, which do not accurately capture cell-level interactions and may introduce noise in their inferences. We envision that as single-cell CCC methods get developed and improved, building hierarchical structures from individual-cell-level CCC is a promising future direction.

Hierarchical clustering on CCC calculated using Scriabin

Hierarchical clustering on CCC calculated using NICHES

3. *Sensitivity of hierarchical clusters: The authors should evaluate the robustness of the detected cluster hierarchies to noise, sparsity, and normalization of the data. Suggestion: Include sensitivity analyses that evaluate these factors and report the robustness of the main biological findings to the assessed variations. Furthermore, the authors should evaluate alternative ways of finding cluster hierarchies (by using other tools designed for this purpose, <https://doi.org/10.1261/rna.078965.121>).*

Response: Thank you for your suggestion. In our updated manuscript, we have added detailed descriptions of how we evaluated the robustness of CrossChatH in the Methods section (Robustness of CrossChatH).

Following your suggestion, we have tested hierarchical clustering of CrossChatH to selection of K in K-nearest neighbor graph construction, noise, sparsity, and normalization, on a PBMC dataset (Supplementary Fig. 11). To evaluate the robustness of hierarchical clustering with respect to K value in KNN graph construction, we ran the method for different K values: 5, 10, 15, 20, 25, and 50. We calculate the multiscale similarity between clustering of different K values and observed that the score between each multiscale clustering is greater than 0.9 most of the time, indicating a high robustness to K value selection. To evaluate the robustness of clusters to noise, we generated 3 scaled Gaussian noise and added them to the gene expression matrix. Specifically, the Gaussian noise is multiplied by three different scales: 10^{-3} , 10^{-4} , and 10^{-5} . The multiscale similarity between the clustering with added Gaussian noise and the clustering with no noise is still very high, indicating a high robustness of our clustering to noise. To evaluate the robustness to dropout, we chose 4 dropout rates: 0.1, 0.2, 0.3, and 0.5. We achieve this by randomly setting 10%, 20%, 30% and 50% of the gene expression levels to 0. The multiscale similarity decreases when dropout rate increases. Finally, to evaluate the robustness to different normalization methods, we tested three popular normalizations, including log-normalization, counts per million, and centered log-ratio transformation. We found that normalization using either log-normalization or counts per million yield similar multiscale clustering results, while normalizing using centered log-ratio transform generates results more different from the other two.

The hierarchical clustering methods in the listed paper produce a dendrogram, and output one scale of clustering, as there is not a way to choose multiple resolutions in their methods. Our method can find robust resolutions automatically at multiple scales.

4. More details should be provided about the methodology. For example:
 a. When building the cell-cell similarity graph based on cosine similarity from the principal components, did the authors use the whole gene expression matrix or just ligands and/or receptors?

Response: Thank you for your comment. The cosine similarity is calculated either based on all genes, or ligands/receptors, based on user interest. For example, in our analysis of PBMC cells of COVID-19 patients, we used all genes as input to obtain a general tree (Fig. 4c). Then we used ligands/receptors as input to get ligand tree and receptor tree

(Fig. 4e). Users can modify the input to be all genes, ligands, or receptors in our pipeline. We have added clarifications as above in the Methods section.

b. The use of CellChat and COMMOT was mentioned for inferring CCC in single-cell and spatial data, but it is not clear how and when these scores were computed and passed to their approach.

Response: Thank you for your comment on improving the clarity of our method. In our updated manuscript, we have added a section, Incorporation of CellChat and COMMOT in Methods, which gives more details on when and how these scores are computed. Once the hierarchical clusters are found using CrossChatT or CrossChatH, we compute interactions between the identified hierarchical clusters using either CellChat (for nonspatial datasets) or COMMOT (for spatial datasets). CellChat takes the clusters and averages the ligands/receptors expression and other information from gene expression, and calculates the interaction scores between clusters. COMMOT takes gene expression and spatial information and calculates the interactions between each pair of cells. Then we aggregate the total interactions for all cells in clusters. Specifically, CellChat deduces significant CCC activity from scRNA-seq data between identified cell groups. It subsequently computes an interaction score using mass action kinetics, which represents the probability of CCC. This process integrates gene expression data with existing knowledge about the interactions among signaling ligands, receptors, and their cofactors. COMMOT constructs cell-cell communication (CCC) networks using predefined ligand-receptor pairs by solving a global optimization problem. This method accounts for higher-order interactions among multiple ligand and receptor species. It introduces collective optimal transport to determine optimal transport plans for all species pairs simultaneously, allowing for interdependent couplings that are not possible with traditional optimal transport.

c. The method section about CCC tree search should be better explained, specifically regarding what is a disjoint or inclusive relationship and how this is included and used by the algorithm. Consider adding a supplementary figure representing how these

relationships are found from the binary values and how the trees are built (explicitly showing connections for a toy example).

Response: Thank you for your suggestion to improve the clarity of CCC tree search method. Following your suggestion, we have added a Supplementary Figure to better illustrate how the trees are built from gene expression input (Supplementary Fig.10). In our updated Methods section (CrossChatT: CCC tree search), we clarified the disjoint or inclusive relationship between ligands/receptors, and improved explanations of our method as follows.

We first binarize the gene expression matrix in order to find the support for each ligand/receptor. Next, we model that two ligands/receptors are connected if their relationship (with respect to their supports) is either disjoint or inclusive. Two ligands/receptors are disjoint of each other if the cells containing them are exclusive. One ligand is a subset of another ligand if the cells expressing ligand 1 also express ligand 2. Thus, we form a gene relationship graph based on the relationship between ligands/receptors. In a tree of interest, all pairs of ligands/receptors are either disjoint or inclusive, thus are connected. Such a tree structure forms a complete graph, where nodes represent ligands/receptors, and every pair of nodes has an edge connecting them. In order to find largest trees of ligands/receptors in which any other ligand/receptor is either intersecting with one ligand/receptor in the tree or is disjoint with any ligand/receptor in the tree, we use Bron-Kerbosch graph search algorithm to find all maximal complete subgraphs. Finally, we check each maximal complete subgraph and extract all individual trees from them.

Illustration of CrossChatT with a toy example

5. More details about the selection of parameters and analysis decisions should be provided. Furthermore, a benchmarking of different strategies and parameters should be provided. For example:

a. The authors should clarify the rationale for using the geometric mean for aggregating p -values of all genes in the ligand or receptor complex. Multiple approaches exist (e.g., <https://www.pnas.org/doi/10.1073/pnas.0406811102>, <https://doi.org/10.1371/journal.pone.0125081>, <https://www.nature.com/articles/s41598-021-86465-y>), so it would be important to indicate why they chose the geometric mean over other approaches. Consider providing a comparison using a couple of different methods to show how they could affect the results.

Response: We apologize for the confusing description of calculating the p -value of ligand or receptor complex in our previous manuscript. Indeed, for each ligand or receptor complex, we use the geometric mean of each ligand or receptor in the complex to represent the ligand (receptor) complex expression, as has been done in CellChat. Instead of aggregating p -values of all genes in the ligand (receptor) complex, we only

calculate the p -value once for the ligand (receptor) complex using its aggregated gene expression. In our updated manuscript, we have modified our previously imprecise description on calculating the p -value for ligand (receptor) complex in method section (Obtain ordered list of specific ligand-receptor pairs).

b. The authors should compare the use of tools other than CellChat and COMMOT for inferring CCC from single-cell and spatial data and evaluate how the hierarchical structures change. See alternative methods here: <https://doi.org/10.1038/s41576-023-00685-8>.

Response: Thank you for your comment. Following your suggestions, we have recalculated CCC using NATMI (PMID: 33024107) (for single-cell data), and Giotto (PMID: 33685491) (for spatial data), and compared CrossChat to them. When analyzing the COVID-19 PBMC data, we can see that the hierarchical cell structures detected by CrossChatH do not change even when we change the CCC calculation method, albeit there are slight differences in the calculated interaction strengths between CellChat and NATMI. When analyzing a spatial 10x Visium dataset of wounded mouse skin, by comparing between COMMOT and Giotto, we can see again that the hierarchical cell structures detected by CrossChatT do not change when the CCC calculation method is changed, albeit there is a slight difference in the interaction strengths between cell groups of trees. Overall, these experiments demonstrate that because CrossChat relies solely on gene expression to detect hierarchical structures, the choice of CCC calculation methods does not impact the hierarchical structures.

a CrossChatH and CellChat: CCL3 - CCR1

CrossChatH and Natmi: CCL3 - CCR1

CrossChatH and CellChat: RETN - TLR4

CrossChatH and Natmi: RETN - TLR4

b CrossChatT and COMMOT Interaction tree 1

CrossChatT and Giotto Interaction tree 1

Interaction tree 2

Interaction tree 2

c. What parameters were used for binarizing the gene expression matrix in the CCC tree search? Different ways of binarizing these values should be provided, showing how this could affect the detection of local hierarchy.

Response: Thank you for your question. Binarization normally uses 0 as default threshold to determine expression of a ligand in a cell. However, we provide functionality so that users can set a higher threshold value.

To experimentally test how different binarization affect the detection of local hierarchy, we removed the smallest positive 0%, 5%, 10%, 15% and 20% entries in normalized gene expression matrix of a mouse embryo skin dataset. As a side note, 20% is reasonably high to reduce noise. We didn't test higher binarization threshold because much higher binarization threshold leads to inaccurate inclusion/disjointness relationship. Then, CrossChatT is used to search for ligand trees and receptor trees. We then compared how different binarization threshold will influence the number of detected ligand/receptor trees and the average number of ligands/receptors in the trees. We found that the number of detected ligand trees decrease as we increase the binarization threshold. The change in number of detected receptor trees, or in average number of ligands/receptors in ligands/receptors trees is not significant with the change of binarization threshold.

6. Implement an automatic approach for finding important structures of LR pairs: While the authors highlight important pairs of clusters with distinct hierarchies and key LR interactions, it is unclear how they selected them. It seems that a certain level of expertise is needed to fully use the tool's capabilities.

Suggestion: The authors should implement an unsupervised way of selecting important hierarchical structures that could be pertinent to a dataset of interest. Alternatively, they should provide guidance on what to focus on for selecting important structures (maybe include a brief section for this).

Response: Thank you for your suggestion. CrossChatH is unsupervised and automatically selects one hierarchical structure in cells based on gene expression. Users just need to input their preference of using all genes or ligands/receptors as input to detecting hierarchical structures. CrossChatH also automatically selects an ordered list

of specific ligand-receptor interactions. We describe this procedure in more detail in the Methods section.

For CrossChatT, following your suggestion, when detecting ligand/receptor trees, we added additional functionality for tree selection by ranking the number of ligands/receptors in ligands/receptors trees. Furthermore, CrossChatT can now detect the pairs of ligand and receptor trees involved in ligand-receptor interactions, using the function `finding_interacting_trees`. The updated functions can be found on our GitHub(<https://github.com/Xinyiw28/CrossChat/blob/main/src/crosschat/CrossChatT.py#L297>). After these changes, users only need to input the gene expression matrix (either raw counts or normalized) to find an ordered set of ligands/receptors trees, and detect the pairs of trees with interactions automatically.

7. For the sake of reproducibility and tool usage, the authors should improve the documentation of CrossChat by including, for example, a readthedocs website. In addition, they should deposit the tool in PyPI or conda to facilitate its installation. Finally, they should incorporate changes related to my other comments into their tutorial, making the tutorials as comprehensive as possible.

Response: Thank you for your suggestions on improving reproducibility of CrossChat package. Following your suggestions, we have done the following to improve documentation of CrossChat.

1. Included a Read The Docs website:

<https://crosschat.readthedocs.io/en/latest/README.html>

2. Deposited the tool in PyPI for installation:

<https://test.pypi.org/project/crosschat/0.0.1/>

Minor comments:

1. The detail on the PBMC dataset simulation (page 6) can be moved to Methods or Supplementary Information to improve readability. Just convey the key points in the Results.

Response: Thank you for your suggestion on improving readability of our paper. Following your suggestion, we have moved the details of the PBMC dataset simulation to the Methods section, and reorganized them in a step-by-step manner.

2. *Provide more intuition on the Markov stability metric and explain the resolution parameter t in the CrossChatH Methods.*

Response: Thank you for your suggestion. In the Methods section (“Random walk on graph and Markov stability”), we have further explained the intuition of Markov stability, and its resolution parameter t : “Intuitively, for a given way of partitioning the graph into communities, Markov stability measures the overall likelihood that a random walker stays inside its community at time t . If we fix the communities, this likelihood will decrease as t grows, since the random walker tends to explore more other communities rather than staying in its own. For each t -value between zero and infinity, we use Louvain community detection algorithm to find communities that maximizes the Markov stability at this t -value. Due to the previous fact that Markov Stability decreases as t increases for fixed communities, we tend to have fewer (and larger) communities when maximizing Markov Stability for bigger t -values, until t is infinity where one community that contains the whole graph is the maximizer for sure. As mentioned above, as t grows, finer clusters will gradually merge into larger clusters and thus forms the hierarchical clustering. In CrossChatH, we use this method to detect cell clusters for different time parameter t . Some of the t -parameters will lead to numerically unstable clusters and we rely on the variational method in the following section to select robust ones.”

3. *The word "multiscale" sounds a bit ambiguous and sometimes hard to follow. From my point of view, it could be better to use something like "varying resolution" or other alternatives instead.*

Response: Thank you for your suggestion. We agree that multiscale may be vague in describing hierarchical structures, or clusters at “varying resolution”. In our revised manuscript, we have modified “multiscale” to “multi-resolution”, “clusters at varying resolutions”, or “hierarchical structures in CCC” to improve clarity.

4. Does the method allow the inclusion of a predefined hierarchy? For example, using the hierarchy of cells seen during development along time (progenitor-precursor-differentiated cell relationship). This should be discussed, and maybe an example could be provided in the tutorials.

Response: This is a great point. Our method allows users to input their own hierarchical clusters as input. An example of such usage has been added in our updated tutorial (https://github.com/Xinyiw28/CrossChat/blob/main/Tutorials/CrossChatH_tutorial.ipynb).

Reviewer #2 (Remarks on code availability):

For the sake of reproducibility and tool usage, the authors should improve the documentation of CrossChat by including, for example, a readthedocs website. In addition, they should deposit the tool in PyPI or conda to facilitate its installation. Finally, they should incorporate changes related to my other comments into their tutorial, making the tutorials as comprehensive as possible.

Response: Thank you for your suggestions on improving the reproducibility of CrossChat package. Following your suggestions, we have done the following to improve documentation of CrossChat.

1. Included a readthedocs website:

<https://crosschat.readthedocs.io/en/latest/README.html>

2. Deposited the tool in PyPI for installation:

<https://test.pypi.org/project/crosschat/0.0.1/>

Reviewer #3 (Remarks to the Author):

This topic is great. To the best of my knowledge, it is indeed the first method to detect hierarchical structures within CCC. The author also demonstrated through the explanation in the “Introduction” section and the analysis of three real data using CrossChat in the “Results” section that hierarchical structures within CCC can indeed bring effective

biological discoveries. The authors also used simulation data for validation, while adapting to single-cell and spatial transcriptome data, giving CrossChat a wider range of application scenarios. CrossChat does have a good appeal to biologists and can also advance algorithm research in the field of intercellular communication. I think that this article can be considered for acceptance after appropriate modifications.

Response: Thank you very much on your appreciation of the novelty and functionalities of CrossChat, and for your insightful suggestions. We have made multiple revisions to the manuscript and believe that these revisions have improved the work. Changes are highlighted in red throughout our updated manuscript. Below are our detailed responses to your suggestions.

Major points:

1. The author only used simulation data to verify the function of detecting specific CCC of CrossChatH, which is only a small part of the CrossChat function. Can the author provide a validation for functions from “CrossChatT”?

Response: Thank you for your suggestion. Following your suggestion, we performed experimental quantitative validation of the CrossChatT in our setting. Specifically, we generate 1000 cells, where each cell expresses 100 genes. We distributed the gene expression so that there are at least five trees as shown in the figure. There are 15 genes involved in these five trees. For each of the remaining 85 genes, we randomly chose 500 cells to express them. We ran the experiment 1000 times. CrossChatT is able to detect all five trees we generated in each trial (Fig. 3e). The algorithm used by CrossChatT to find all maximal complete subgraphs is Bron-Kerbosch. This algorithm is an enumeration algorithm, and has been demonstrated to be capable of finding all maximal complete subgraphs. Finding hierarchical ligands/receptors trees is exactly as finding maximal complete subgraphs in the graph of ligands/receptors that we constructed (see Methods: CrossChatT: CCC Tree search). Thus, as this algorithm is exact in detecting all maximal complete subgraphs, it is capable of detecting all existing hierarchical ligands/receptors trees.

2. A more detailed description is needed on how CrossChat uses spatial information, especially regarding whether CrossChat H-S can effectively utilize spatial information by “concatenating to the PCA embeddings of cells”. If there is a precedent for this approach, please include appropriate references. Alternatively, can the author compare the results with and without spatial information based on spatial transcriptomic data to verify that CrossChat effectively utilizes spatial information?

Response: Thank you for your suggestion about improving the clarity of CrossChat’s spatial incorporation approach. This is our newly designed method to integrate spatial information. CrossChatH-S first obtains the PCA embeddings of cells based on gene expression or ligands/receptors expression, depending on users’ preference. Then, we scaled the spatial position of spatial spots in both directions to be within the range from 0 to 1. For each cell, the two vectors (PCA embedding and scaled spatial position) are then concatenated to form a new vector, which is used to calculate the similarity of cells and construct the KNN graph on cells. In our updated manuscript, we expanded the “Incorporating spatial information” section in Methods to incorporate these descriptions

We experimentally compared the results with and without spatial information based on spatial transcriptomic data of mouse embryo. When incorporating hierarchical clustering result with spatial incorporation, we found that detected clusters are spatially adjacent to each other compared to the clustering without spatial incorporation. For example, we can see that neurons (MSCN+Forebrain) cannot be separated into different spatial regions. With spatial incorporation, the neurons are further segmented into multiple spatial regions. We also observed that multiscale clusters that utilize spatial information has a higher spatial neighborhood enrichment score, indicating better spatial coherence. In our updated manuscript, we have added the experimental comparison between hierarchical clustering with and without spatial incorporation in the Results section (CrossChatS reveals hierarchical clusters and ligand-receptor interactions in spatial datasets).

CrossChatH with no spatial incorporation and neighborhood enrichment scores

3. For CrossChatT-S, the author uses the following settings: "For any ligand or receptor, we restrict its support such that only ligands or receptors whose nearby spots express its receptors/ligands are kept in its support". However, there are significant differences in resolution and capture ability among different spatial sequencing methods. For spatial transcriptomic data with very sparse gene expression, only considering adjacent spots will result in only a small number of LR pairs being detected. Can the author set a

parameter so that users can choose to include CCC within which spatial distance range based on their own data?

Response: This is a great point. Yes, users can choose their own spatial range based on their own data in order to adapt the differences in spatial sequencing methods. In our revised manuscript, we have highlighted this point in “Incorporating spatial information” section in Methods.

Minor points:

1. The caption of Figure 2 includes a, b, and c, but only a and b are present in the image. Maybe both a and b in the caption correspond to Figure 2a.

Response: Thank you for your comment. We have modified our figure captions of Figure 2 to include only a and b.

2. In the leftmost subgraph of Figure 2b, the annotation for the blue subgroup is missing and should match the red, yellow, and blue subgroups.

Response: Thank you for your suggestion. We have added the annotations that match each group accordingly in the updated Figure.

3. Is there a corresponding relationship between (scale 2)-(cluster1-8) in Figure 1a and each leaf node (C4-C11) in Figure 1d? If so, maybe the authors should consider naming each leaf node in Figure 1d as SC1-SC8.

Response: Thank you for your suggestion on improving the readability of our figures. We have followed your suggestion and modified the annotations of each leaf node accordingly in Figure 1d.

4. Consider moving some of the content from the section "Validation of CrossChat using

simulated dataset and COVID-19 dataset" into the Methods section and re-organizing it in a step-by-step manner.

Response: Thank you for your suggestion on improving the readability of our manuscript. Following your suggestion, we have moved the details about the PBMC dataset simulation to Methods section, and reorganized them in a step-by-step manner.

5. A preprocessed toy data that meets the CrossChat input requirements should be provided in the tutorial of Github, or a toy data download link should be provided.

Response: Thank you for your suggestion. We have uploaded all data from tutorials in Zenodo. People can download the input data needed for tutorials from this link: <https://zenodo.org/records/13186437>. We have also updated our tutorial page to include link:https://github.com/Xinyiw28/CrossChat/blob/main/Tutorials/CrossChatH_tutorial.ipynb)

6. The name of each node, at least the nodes connected by the edges of CCC, should be provided in the output image of the "ccH_obj.Draw_CCC_LR" function.

Response: Thank you for your suggestion. We have updated the function of "ccH_obj.Draw_CCC_LR" to include legends of nodes. Specifically, we equipped each detected multiscale cluster with a label, and prepared a heatmap which visualizes the similarity between each cluster and human-labeled cell type annotations.

REVIEWER COMMENTS

Reviewer #1 (Remarks to the Author):

Thank you to the authors for their comprehensive responses to my previous concerns. All of those concerns have been adequately addressed in the revised manuscript. I have only two minor suggestions that could be incorporated by minor text editing.

Minor comments:

1. Several traditional CCC inference methods have been widely used in single-cell and spatial transcriptomics studies, such as CellChat and CellphoneDB. Could the authors provide a short guideline in the Discussion section for selecting their proposed CrossChat method over traditional approaches? Under what circumstances might CrossChat be particularly advantageous?

Thank you for your suggestion to emphasize the advantages of CrossChat over existing packages. Following your suggestion, we have added the following in our updated manuscript: In general, CrossChat is particularly advantageous over other existing detection methods in the following scenarios: 1) Analysis of CCC between cell groups at multiple clustering resolutions, rather than at a single scale 2) Analysis the structures within cell-cell communications that are specific to signal ligands/receptors, which are independent of predefined cell type annotations.

We have updated Supplementary Figure 8a–b to show a comparison between existing single-resolution CCC methods (including CellChat and CellphoneDB) and our multi-resolution method CrossChat.

2. Supplementary Fig. 1 referenced on Line 236, 248, 275, 285 and 288 should be Supplementary Fig. 3.

Thank you for your comment. We have updated in our manuscript to ensure the correct reference to this Supplementary Figure.

Reviewer #2 (Remarks to the Author):

I appreciate the authors' efforts in addressing the previous review comments and improving their manuscript. The changes have significantly enhanced the quality of the work. However, there are a few important points that should be addressed before publication:

1. Many analyses performed in response to reviewer comments should be incorporated into the manuscript since they could be informative to readers wondering about similar ideas. These should be presented as supplementary figures or notes and referenced in the main text. Specifically:

- a) Figures responding to reviewer #2's point #2 (single-cell tools + hierarchical clustering on CCC scores)
- b) Figures for reviewer #2's point #5b (including other CCC tools for predicting communication used as input of CrossChat)
- c) Figures for reviewer #2's point #5c (using different thresholds for binarizing gene expression)
- d) Any other relevant analyses not currently included

Please integrate these analyses and discuss or mention them, even if briefly, to provide readers with a comprehensive understanding of the methodology and its robustness.

Thank you very much for these valuable suggestions. We have now included these missing pieces into our updated manuscript (line 168-173, line 487-518, line 536-541), and added the figures as new Supplementary Figures (Supplementary Fig. 1, 8, 10-12).

2. While the authors have addressed the use of single-cell tools and clustering (reviewer #2's point #2), the intention of my original comment was slightly different. My original thought was about using this alternative approach as a baseline. This approach would through using tools like NICHES or Scriabin to infer a CCC matrix of single-cell sender-receiver pairs (rows/columns) by all LR pairs (columns/rows). Then, perform traditional hierarchical clustering, instead of CrossChat approach, using the whole vector per cell pair (i.e., all scores of LR pairs in a pair of cells). You can run for example the hierarchical clustering employed to generate dendrograms in scipy or seaborn clustermaps. Finally, compare the resulting substructures with those obtained from CrossChat. Please conduct this analysis and discuss how the results compare to CrossChat's.

Thank you for your insightful suggestion. Following your advice, we attempted to explore the clustering of cell pairs with respect to CCC, using hierarchical clustering on a COVID-19 dataset (PMID: 33657410). Specifically, we used fcluster and linkage functions from SciPy's `scipy.cluster.hierarchy` module to perform hierarchical clustering, and used `dendrogram` function from `scipy.cluster.hierarchy` to generate dendrogram visualizations. While this approach is conceptually intriguing, it presents significant challenges when applied to scRNA-seq datasets for two main reasons:

First, although our test dataset contains only 3,000 cells, it generates 9 million cell pairs, making hierarchical clustering computationally infeasible due to both memory and time complexity constraints. Specifically, the hierarchical clustering function stores a condensed distance matrix to keep track of pairwise distances between data points. For 9 million data points, this matrix would require storage for approximately 81 trillion distances, which makes it infeasible on standard hardware in terms of required memory. Furthermore, the hierarchical clustering has a time complexity of $O(n^2)$, which also becomes prohibitively expensive when scaling to 9 million data points. Given that many

modern scRNA-seq datasets contain millions of cells, far exceeding our sample size of 3,000, such an analysis becomes even more impractical with currently available computational resources.

To gain at least a basic understanding of how traditional hierarchical clustering might work on cell pairs, we reduced the dataset to 100 cells, resulting in 10,000 cell pairs—small enough to meet computational constraints. We generated a heatmap that illustrates the overlap of sender-receiver cell groups across ten clusters of cell pairs. However, the cell-pair structures proved to be less interpretable than the original cell groups, largely because different clusters of cell pairs could include identical sets of cells (simply paired differently), making it difficult to visually distinguish meaningful differences between the clusters. For example, in the middle figure below (part of Supplementary Figure 12), the Jaccard similarity of sender/receiver cells in cell pair cluster 5 and 9 is 0.98, indicating that the group of senders and receivers are indeed almost identical even though they're in two different clusters of cell-pairs. To ensure the robustness of this finding, we also performed subsampling and averaged the results over 20 subsamples. We found that the average number of pairs with similarity higher than 0.5 is 11.6, indicating that many of the clusters of cell pairs have a high overlap.

c Dendrogram clustering of 10000 cell pairs

d Similarity of sender/receiver cells of cell pair clusters in 1 trial

3. For reviewer #2's point #5b: The current comparison with NATMI and Giotto is a good start, but not conclusive to say that CrossChat's result do not change across CCC tools given the small number of tools employed here. I recommend including at least a couple of extra tool in each case, covering distinct strategies to infer CCC:

- a) For single-cell data: LIANA (using aggregated rank across multiple methods) and SingleCellSignalR (with regularized score)
- b) For spatial data: SpatialDM (based on auto-correlation Moran's I), DeepLinc, and spaCI (both based on deep learning but with different loss functions)

Please expand the comparison to include these additional tools and discuss the implications of the results.

Thank you for your comment. Following your suggestions, in addition to NATMI (PMID: 33024107) and Giotto (PMID:33685491), we have recalculated CCC using LIANA (PMID: 39223377), SingleCellSignalR (PMID: 32196115) for scRNA-seq data, SpatialDM (PMID: 37414760) and spaCI (PMID: 36545790) for spatial data. In spaCI, we used its calculation which is the multiplication of average ligand/receptor values as its final CCC score. Since DeepLinc (PMID: 35659722) only infer a general interaction strength between cell groups, instead of ligand-receptor interactions that rely on cell-cell communication databases, which is the focus of CrossChat, thus it cannot be incorporated with CrossChat.

In our analysis of COVID-19 PBMC data (PMID: 33657410) using CrossChatH, we found that the hierarchical cell structures identified remain consistent across various CCC calculation methods. However, interaction strengths exhibit minor variations due to the different scoring functions employed by each method. Similarly, when examining a spatial 10x Visium dataset of wounded mouse skin and comparing the results generated by

COMMOT, Giotto, SpatialDM, and spaCI, it is evident that the hierarchical structures detected by CrossChatT are stable despite changes in the CCC calculation approach. The only variation noted was in the interaction strengths among cell groups within these structures. This consistency underscores a key aspect of CrossChat's functionality: it initially identifies hierarchical structures based on only on ligand/receptor expression, independent of the CCC calculation method used. Thus, while changes in CCC calculation influence the interaction strengths, they do not alter the underlying detected structures.

a CrossChatH and CellChat: CCL3 - CCR1

CrossChatH and CellChat: RETN - TLR4

b CrossChatH and Natmi: CCL3 - CCR1

CrossChatH and Natmi: RETN - TLR4

c CrossChatH and LIANA: CCL3 - CCR1

CrossChatH and LIANA: RETN - TLR4

d CrossChatH and SingleCellSignalR: CCL3 - CCR1

CrossChatH and SingleCellSignalR: RETN - TLR4

a CrossChatT and COMMOT

b CrossChatT and Giotto

c CrossChatT and SpatialDM

d CrossChatT and spaCI

4. While the ReadtheDocs documentation provides useful tutorials, it lacks detailed API documentation. Please add comprehensive API documentation, including docstrings and information about input/output parameters for each function in the codebase. This will greatly assist users in understanding the tool's functionality at each step.

Thank you for your comment. In our updated ReadtheDocs website, we added docstrings and information about input/output parameters for each function to improve its readability and accessibility to users: <https://crosschat.readthedocs.io/en/latest/index.html>.

Reviewer #3 (Remarks to the Author):

My concerns have been addressed in the revisions.